



# Quantifying wind plant blockage under stable atmospheric conditions

Miguel Sanchez Gomez[1], Julie K. Lundquist[1,2], Jeffrey D. Mirocha[3], Robert S. Arthur[3], and Domingo Muñoz-Esparza[4]

[1]Department of Atmospheric and Oceanic Sciences, University of Colorado Boulder, Boulder, Colorado, 80309-0311, United States
[2]National Renewable Energy Laboratory, Golden, Colorado, 80401, United States
[3]Lawrence Livermore National Laboratory, Livermore, California 94550, United States
[4]National Center for Atmospheric Research, Boulder, Colorado 80301, United States

**Correspondence:** Miguel Sanchez Gomez (misa5952@colorado.edu)

**Abstract.** Wind plant blockage reduces the wind velocity upwind undermining turbine performance for the first row of the plant. We assess how atmospheric stability modifies the induction zone of a wind plant in flat terrain. We also explore different approaches to quantifying the magnitude and extent of the induction zone from field-like observations. To investigate the influence from atmospheric stability, we compare simulations of two stable boundary layers using the Weather Research and 5 Forecasting model in large-eddy simulation mode, representing wind turbines using the generalized actuator disk approach. We find a faster cooling rate at the surface, which produces a stronger stably stratified boundary layer, amplifies the induction zone of both an isolated turbine and of a large wind plant. A statistical analysis on the hub-height wind speed field shows wind slowdowns only extend far upstream (up to 15D) of a wind plant in strong stable boundary layers. To evaluate different ways of measuring wind plant blockage from field-like observations, we consider various ways of estimating the freestream velocity 10 upstream of the plant. Sampling a large area upstream is the most accurate approach to estimating the freestream conditions, and thus of measuring the blockage effect. Also, the choice of sampling method may induce errors of the same order as the velocity deficit in the induction zone.

*Copyright statement.* This work was authored [in part] by the National Renewable Energy Laboratory, operated by Alliance for Sustainable Energy, LLC, for the U.S. Department of Energy (DOE) under Contract No. DE-AC36-08GO28308. JDM and RSA contributed under the 15 auspices of the U.S. Department of Energy (DOE) by Lawrence Livermore National Laboratory, under contract DE-AC52-07NA27344. Funding provided by the U.S. Department of Energy Office of Energy Efficiency and Renewable Energy Wind Energy Technologies Office. The views expressed in the article do not necessarily represent the views of the DOE or the U.S. Government. The U.S. Government retains and the publisher, by accepting the article for publication, acknowledges that the U.S. Government retains a nonexclusive, paid-up, irrevocable, worldwide license to publish or reproduce the published form of this work, or allow others to do so, for U.S. Government 20 purposes





## 1 Introduction

Wind turbines extract kinetic energy from the wind, thereby reducing its velocity downstream. As a result, turbines downstream experience lower wind speeds with higher turbulence and produce less power, an effect known as wake loss. However, wake losses are not the only way in which a wind plant's power production is reduced. Winds also decelerate upstream due to a

positive pressure gradient that forms from turbines obstructing and extracting momentum from the flow. This effect is known as wind turbine blockage. And, when multiple turbines are combined into an array, the upstream wind speed decrease is larger compared to that of a turbine in isolation. This array effect is known as wind plant blockage. Generally, wake effects are accounted for in power forecasting methods that employ deterministic tools (Wang et al., 2011; Bleeg et al., 2018). However, upstream wind plant blockage is usually neglected, resulting in lower-than-forecasted energy predictions and financial losses

for wind plant operators (Ørsted, 2019).

Though wind speed deficits from wakes are large (∼10%), wind plant blockage produces wind speed deficits of ∼1% over a wide area upstream, making it much more difficult to quantify. This region of reduced wind speeds is called the induction zone of the wind plant. Numerical and experimental studies show the extent and magnitude of the induction zone can vary significantly depending on the size and layout of the wind plant, atmospheric conditions, wind turbine characteristics, and

wind speed (Allaerts and Meyers, 2017, 2018; Bleeg et al., 2018; Wu and Porté-Agel, 2017; Segalini and Dahlberg, 2019; Schneemann et al., 2021). Whereas some studies show slowdowns up to ∼30D upstream (Wu and Porté-Agel, 2017; Segalini and Dahlberg, 2019; Schneemann et al., 2021; Bleeg et al., 2018), others show slowdowns that extend up to ∼80D upstream of first row of the wind plant (Allaerts and Meyers, 2018; Wu and Porté-Agel, 2017). The spread in the wind speed deficit observed just upstream of the plant is even larger. On one hand, several simulations found wind decelerations larger than 10%

at a distance of 2D upstream of the first row of turbines (Allaerts and Meyers, 2018; Wu and Porté-Agel, 2017). On the other hand, experimental results and most numerical simulations suggest wind speed slowdowns that are one order of magnitude smaller at a distance of 2D upstream of the first row of turbines (Bleeg et al., 2018; Wu and Porté-Agel, 2017; Segalini and Dahlberg, 2019; Schneemann et al., 2021). The large spread in the results primarily comes from simulations where the wind plant triggers gravity waves in the domain. The large and uncertain extent of the induction zone, and the small wind speed

deficits detected herein make blockage difficult to quantify, especially in experimental setups.

Numerical simulations and experimental studies quantify the induction zone in different ways, adding uncertainty to the comparison of their results. To estimate the blockage effect, numerical studies usually run two set of simulations, one with and another without the wind turbines in the domain. The difference in the wind speed field between both simulations brings to light the effects from the wind plant and allows quantification of the induction zone. For experimental setups, however, there

is no accepted methodology for measuring the blockage effect. Generally, experimental studies employ a freestream velocity that characterizes the flow to estimate the blockage effect. Wind tunnel experiments estimate the freestream velocity using point measurements far upstream of the wind plant (Segalini and Dahlberg, 2019). Various methodologies have been used to quantify blockage in operational wind plants. Bleeg et al. (2018) employed a statistical approach to differentiate between wind speed point measurements before and after several onshore wind plants were deployed, finding that these induce lower





wind speeds upstream (∼3% deficits). Schneemann et al. (2021), on the other hand, were able to quantify the magnitude of upstream blockage in an operational offshore wind plant using scanning lidar measurements. Schneemann et al. (2021) estimate a freestream velocity for each scan using the mean wind speed across the whole sampling area, which includes the induction zone region. Still, there is not an accepted methodology for defining the unperturbed conditions upstream of wind plants, and hence their blockage effect.

Just as wind speed and atmospheric stability affect turbine power production (Wharton and Lundquist, 2012; Vanderwende and Lundquist, 2012; St. Martin et al., 2016) and wake structure (Bodini et al., 2017; Rhodes and Lundquist, 2013), the induction zone of a wind plant is modified with atmospheric static stability (Allaerts and Meyers, 2018; Schneemann et al., 2021; Wu and Porté-Agel, 2017). Scanning lidar observations at a 400 MW offshore wind plant show upstream blockage occurring during stable conditions and winds between cut-in and rated speed, where the turbines are operating at high thrust
coefficients (Schneemann et al., 2021). They did not find wind slowdowns for other wind speed regimes or atmospheric stability conditions, either suggesting blockage is indeed not existent or that the highly averaged wind speed fields obscured the presence of this phenomenon. Numerical simulations have also shown how blockage is modified by atmospheric conditions. Large-eddy simulations (LES) of a conditionally neutral boundary layer illustrate how the induction zone of a large wind plant changes with free-atmosphere stratification (Wu and Porté-Agel, 2017). Wu and Porté-Agel (2017) suggest the blockage effect is highly
amplified by gravity waves propagating upstream of the plant in subcritical flows ($Fr < 1$). Similarly, Allaerts and Meyers (2018) analyze the blockage effect using LES of stable boundary layers and attribute the flow deceleration to the excitation of gravity waves by the wind plant. Nonetheless, these simulation studies either consider a conditionally neutral boundary layer rather than a stable surface layer, or infinitely large wind plants that overpredict the blockage effect experienced by real wind plants.

Here, we assess how atmospheric stability may amplify upstream blockage, and evaluate different methods of quantifying this phenomenon in mid-sized wind turbine clusters. Specifically, we investigate (1) if a stronger stably stratified boundary layer amplifies the wind speed deficit upstream of a large wind plant, and (2) different methodologies for defining the freestream velocity used to quantify the blockage effect in experimental setups. To evaluate the impact from atmospheric stability, we simulate two distinct stable boundary layers forced with the same geostrophic wind speed. For each case, we run one simulation
with the wind plant and one without to isolate the blockage effect. Furthermore, we run a set of simulations for a single turbine for each atmospheric condition to have a baseline blockage effect for comparison.

The remainder of the paper is structured as follows. We describe our simulation setup and cases in Sect. 2. We show how the blockage effect is modified by atmospheric static stability in Sect. 3. In Sec. 4 we consider the evolution of the boundary layer throughout the domain. In Sect. 5, we present the spatial variability of the velocity upstream of the wind plant and how
this may obscure the induction zone. Wind turbine power production for the first row of the wind plant is evaluated in Sect. 6. Finally, Sect. 7 provides a summary of our findings and suggests future work to further improve our understanding of the wind plant blockage effect.



**Table 1.** Simulation setup including domain size, horizontal resolution, vertical resolution at the surface, and whether or not the Cell-Perturbation Method was activated in the domains. WP indicates the domain for simulations with the wind plant, and ST corresponds to the domain for simulations with a single turbine.

| Domain | $L_x$ [m] | $L_y$ [m] | $\Delta x, \Delta y$ [m] | $\Delta z$ [m] | $n_x, n_y, n_z$ | Cell Perts. |
|---|---|---|---|---|---|---|
| Parent | 30450 | 30450 | 70 | 5 | 436, 436, 67 | No |
| Nest (WP) | 12110 | 11690 | 7 | 5 | 1730, 1670, 67 | Yes |
| Nest (ST) | 7000 | 2520 | 7 | 5 | 1000, 360, 67 | Yes |

## 2 Methodology

### 2.1 Large-eddy simulation setup

We perform Large-Eddy Simulations of wind plants under stable atmospheric conditions using the Weather Research and Forecasting (WRF) model v4.1.5 (Skamarock et al., 2019) with turbines modeled using a generalized actuator disk (GAD) approach (Mirocha et al., 2014). WRF is a fully compressible, non-hydrostatic model that solves the Navier-Stokes and thermodynamic equations for large-Reynolds number fluids (no viscosity or thermal conductivity). WRF uses an Arakawa-C grid staggering in the horizontal and a terrain-following hydrostatic-pressure vertical coordinate. Equations are integrated in time

using a 3rd-order Runge-Kutta scheme, with smaller time step for acoustic and gravity-wave modes. The advection terms are spatially discretized using an even/odd-order numerical scheme.

We use a two-domain configuration with flat terrain to evaluate the blockage effect from wind plants. A periodic LES domain provides the boundary conditions for a nested LES domain via one-way nesting. Horizontal grid spacing for the parent domain is $\Delta x = \Delta y = 70$ m, and for the nested domain is $\Delta x = \Delta y = 7$ m. Both domains share the same vertical resolution, set to

$\Delta z \approx 5$ m below 160 m, then increasing linearly to $\Delta z \approx 80$ m at a height of 1000 m, and finally to $\Delta z \approx 200$ m at the domain top (2500 m). We use the same domain characteristics, but a smaller nested domain size to evaluate the blockage effect for an isolated wind turbine. A summary of the LES domains is in Table 1.

All simulations are initialized dry, with zero latent heat flux. No cloud, radiation, or land surface models are used in the LES domains. To properly simulate the stable boundary layers, we prescribe a cooling rate rather than a heat flux at the surface

(Basu et al., 2008). We include a Rayleigh damping layer with a coefficient of 0.003 s$^{-1}$ in the upper 500 m of each domain to avoid wave reflection from the model top. Surface boundary conditions are specified using Monin-Obukhov similarity theory with a surface roughness of $z_0 = 0.1$ m. We use the nonlinear backscatter and anisotropy (NBA) model with TKE-based stress terms from Kosović (1997), implemented in WRF by Mirocha et al. (2010), to parameterize sub-gridscale (SGS) fluxes of momentum and heat.

Spin-up time for the parent domain varies with the simulated atmospheric condition. Spin-up of the parent domain is complete when the hub-height wind direction upstream of the plant remains nearly constant in time and is from the west. Simu-





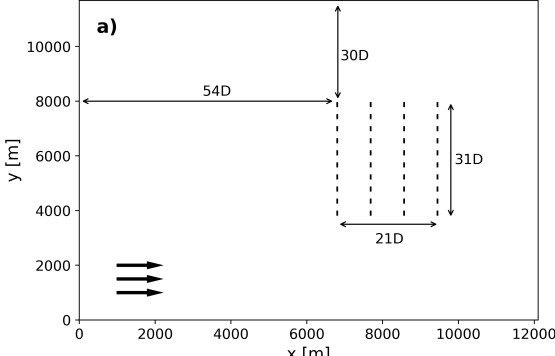

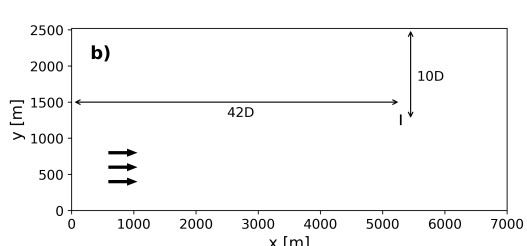

**Figure 1.** Wind plant (a) and single turbine (b) layout within nested domain. The wind direction is from the west.

lations with the faster cooling rate (-0.5 K h$^{-1}$) spin up for 13 hours and 30 minutes, while the simulation with the slower cooling rate (-0.3 K h$^{-1}$) spins-up in 9 hours and 30 minutes. The single-turbine simulations share the same spin-up time as the equivalent simulations that contain the wind plant. We trigger turbulent motions in the nested LES domain using the

cell-perturbation method (CPM) from Muñoz-Esparza et al. (2014) to reduce the fetch required for a fully turbulent flow to develop. The inner domain runs for just over one hour, from which the first 15-30 minutes are discarded (depending on the atmospheric condition) to allow for turbulence to propagate throughout the entire domain. Instantaneous velocity and potential temperature fields are saved every 10 seconds.

    We simulate the wind turbines exclusively in the inner domain using the generalized actuator disk implemented by Mirocha

et al. (2014) and modified by Aitken et al. (2014) and Arthur et al. (2020). The NREL 5MW wind turbine has a hub-height of 90 m, a rotor diameter D of 126 m, cut-in speed at 3 m s$^{-1}$, rated speed at 11.4 m s$^{-1}$, and cut-out speed at 25 m s$^{-1}$. The wind plant and single turbine layout within the domain for our simulations are shown in Figure 1. The wind plant is located more than 30D downstream of the fully turbulent region of the nested domain to allow the induction zone to form within a fully turbulent flow. Similarly, for the single turbine simulations, the turbine is located 18D downstream of the fully turbulent

region of the domain. Our wind plant has an aspect ratio ∼3/2 to amplify the blockage effect as suggested by Allaerts and Meyers (2019). Segalini and Dahlberg (2019) found the blockage effect remains nearly constant when the wind plant has three or more rows, thus we include four turbine rows in our plant. Further, we constrain wind turbine spacing to 7D and 3.5D in the streamwise and cross-stream directions, respectively, for comparison with other simulation studies (Wu and Porté-Agel, 2017; Allaerts and Meyers, 2017, 2018).

## 2.2   Simulated cases

We simulate two different boundary layers to evaluate how blockage varies with static stability (Figure 2). We initialize our simulations with a uniform potential temperature profile $\theta = 300$ K up to $z = 1000$ m, a capping inversion from 1000 m $< z <$ 1200 m with $d\theta/dz = 0.01$ K m$^{-1}$, and we specify $d\theta/dz = 0.001$ K m$^{-1}$ in the troposphere aloft. We consider the effect of



**Table 2.** List of cases and main characteristics for the stable boundary layer simulations. The hub-height wind speed ($U_h$), and inversion height ($z_i$) are the mean values for the simulations. Both cases have the same roughness length $z_0$ at the surface.

| Case | $U_g$ [m s$^{-1}$] | $\dot{T}$ [K h$^{-1}$] | $U_h$ [m s$^{-1}$] | $z_i$ [m] | $z_0$ [m] |
|------|------|------|------|------|------|
| U12-C0.5 | 12 | -0.5 | 8.15 | 170 | 0.1 |
| U12-C0.3 | 12 | -0.3 | 9.44 | 230 | 0.1 |

static stability by forcing the boundary layer with different cooling rates $\dot{T}$ = -0.3, -0.5 K h$^{-1}$ at the surface for the U12-C0.3
and U12-C0.5 cases, respectively. Table 2 contains the main parameters describing the stable boundary layers simulated herein.
The boundary layer height is estimated as the local maximum in $d\theta/dz$ below the capping inversion. The mean boundary layer
height is 170 m for the faster cooling rate case, and 230 m for the slower cooling rate case.

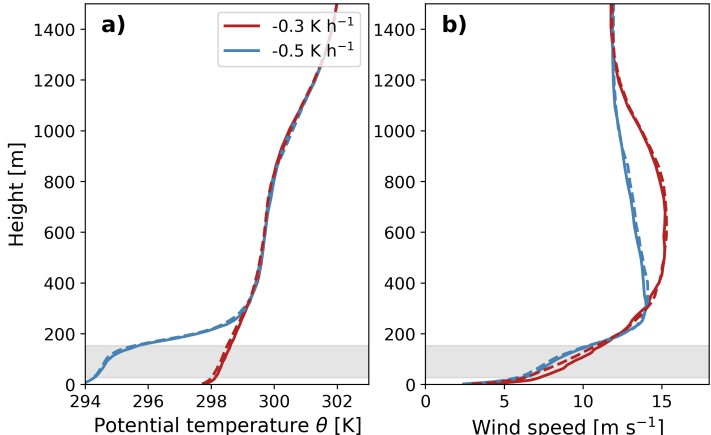

**Figure 2.** Potential temperature (a) and wind speed (b) profiles at $x = 4500$ m and $y = 5840$ m averaged in time for the simulations without
the wind turbines. The dashed colored lines in the plots show the profiles for the simulations with a single wind turbine. The turbine rotor
layer corresponds to the grey shaded region.

## 2.3 Turbulence generation in stable boundary layers

We reduce the fetch required to develop three-dimensional turbulence in the nested LES domain using the cell-perturbation
method (CPM) from Muñoz-Esparza et al. (2014, 2015). The CPM adds random perturbations to the potential temperature
field at the outer eight grid cells of the lateral domain boundaries to instigate three-dimensional turbulent motions below the
capping inversion. We calculate the optimum perturbation amplitudes $\tilde{\theta}_{pm}$ using $Ec = U_g^2/c_p\tilde{\theta}_{pm} = 0.2$, and the time step $t_p$
in between perturbations using $\Gamma = t_p U_1/d_c = 1.15$, as recommended by Muñoz-Esparza et al. (2015). The geostrophic wind
speed ($U_g = 12$ m s$^{-1}$) and the diagonal of the grid cell ($d_c$ = 11.1 m) remain constant for both stability cases. The specific heat





capacity at constant pressure $c_p$ for air in standard conditions is 1006 J kg$^{-1}$ K$^{-1}$. The horizontal wind speed from the parent LES domain at the first vertical level $U_1$ is 2.92 and 2.79 m s$^{-1}$ for U12-C0.3 and U12-C0.5, respectively.

We determine turbulence has propagated throughout the entire domain using the variance of the vertical velocity, which is calculated using 5 minute time windows and averaged along the $y-$direction. The flow becomes fully turbulent at $x = 3000$ m (Figure 3). We discard simulation results upstream of the fully turbulent region of the domain. The vertical velocity variance approximates a quasi-stationary flow 20 - 25 minutes after initializing the nested domain for the U12-C0.3 case (Figure 3). Turbulence in the nested domain takes slightly longer, between 25 - 30 minutes, to propagate throughout the entire domain for the U12-C0.5 case (not shown).

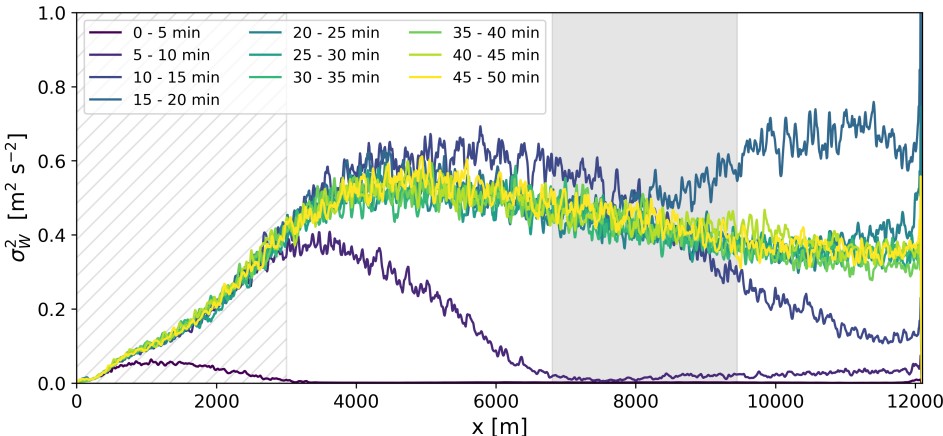

**Figure 3.** Evolution in space and time of the vertical velocity variance for the U12-C0.3 simulation without the wind turbines. The grey shaded region shows the location of the wind plant in the domain. The hatched region shows the area of the domain where turbulence is spinning up.

## 3 The induction zone of a wind plant

We isolate the effect of wind turbines in the flow by running two sets of simulations for each atmospheric condition, one set with and another set without the GAD parameterization. Further, to remove stochastic turbulent motions resulting from CPM, we average the instantaneous 10-s velocity fields output by WRF over the 40 minutes of simulation time. We also perform spatial averaging of the velocity field to extract the effect of each turbine in the flow. In such a way, we only consider the time-averaged hub-height velocity across the domain upstream and downstream of each turbine rotor (Figure 4). Following this approach, we estimate the normalized velocity deficit along the $x-$direction as

$$\hat{\overline{U}}_{def} = \frac{\langle\overline{U}\rangle_{w/} - \langle\overline{U}\rangle_{w/o}}{\langle\overline{U}\rangle_{w/o}}, \tag{1}$$





where $\langle\overline{U}\rangle_{w/}$ and $\langle\overline{U}\rangle_{w/o}$ are the time- and spatially-averaged velocity field for the simulations with and without the wind turbines, respectively. We only consider the velocity field upstream and downstream of each turbine rotor (hatched regions in Figure 4) to compare the velocity deficit of the plant with that of an isolated turbine. Note that in Eq. 1 and what follows, an overbar denotes time averaging, angle brackets denote spatial averaging, and a hat denotes a normalized quantity.

165     We evaluate the statistical significance of the velocity deficit upstream of the first row of turbines in the plant. The velocity at each grid point is assumed to be represented by a normal distribution. For the statistical analysis, we consider the $y-$averaged velocity over the hatched region. The $z-$statistic of the difference of means at each $x-$distance upstream is calculated as

$$z = \frac{\langle\overline{U}\rangle_{w/} - \langle\overline{U}\rangle_{w/o}}{\sqrt{\dfrac{\sigma_{w/}^2}{N_{w/}^*} + \dfrac{\sigma_{w/o}^2}{N_{w/o}^*}}}, \tag{2}$$

where $\sigma_{w/}$ and $\sigma_{w/o}$ are the variance of the velocity field in the simulations with and without the turbines, respectively. The
170     number of independent samples at each distance upstream $N^* = N\dfrac{1-\rho}{1+\rho}$ is estimated using the total sample size $N$ and the lag-1 autocorrelation $\rho$.

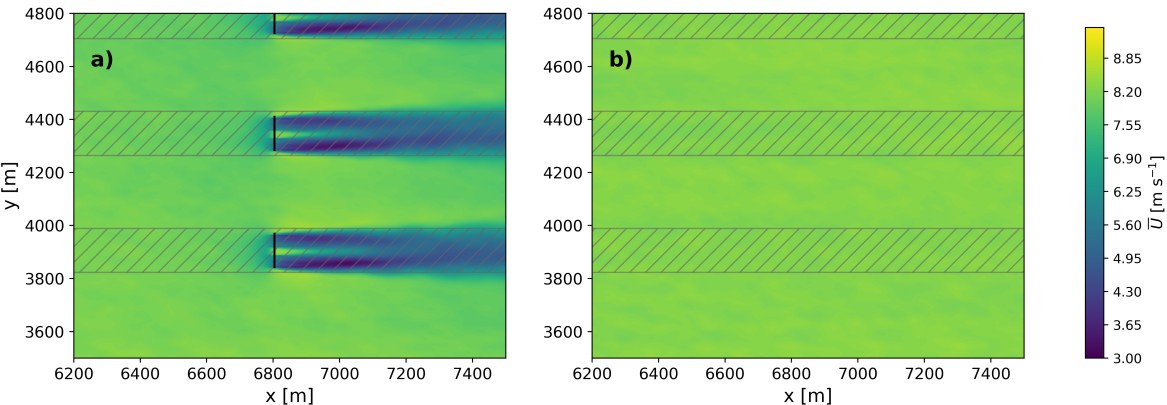

**Figure 4.** Time-averaged hub-height horizontal velocity contours for the U12-C0.5 simulations with (a) and without (b) the GAD parameterization over the same region of the domain. The solid black lines represent the wind turbines in the domain. The hatched region marks the areas considered for the spatial averaging of the velocity field.

Considering the normalized velocity deficit along the $x-$direction for the wind plant and single turbine demonstrates that stronger stable stratification amplifies upstream blockage (Figure 5). While an isolated turbine can influence hub-height winds in a statistically significant ($\alpha = 0.05$) manner up to 5D upstream for the U12-C0.5 case, winds are only statistically different
175     from the winds in the no-turbine simulation up to 1D upstream for the U12-C0.3 case (empty circles in Figure 5). Likewise, hub-height wind speed upstream of the plant ($x < -2.5D$) is statistically different ($\alpha = 0.05$) from the wind speed in the no-turbine simulation only for the U12-C0.5 case (solid circles in Figure 5). The wind plant modifies the flow in a statistically significant manner up to 15D upstream for the U12-C0.5 case, and only 2D upstream for the U12-C0.3 case. Therefore, winds far upstream of a wind plant in a weakly stratified boundary layer are not significantly modified by the plant's blockage.



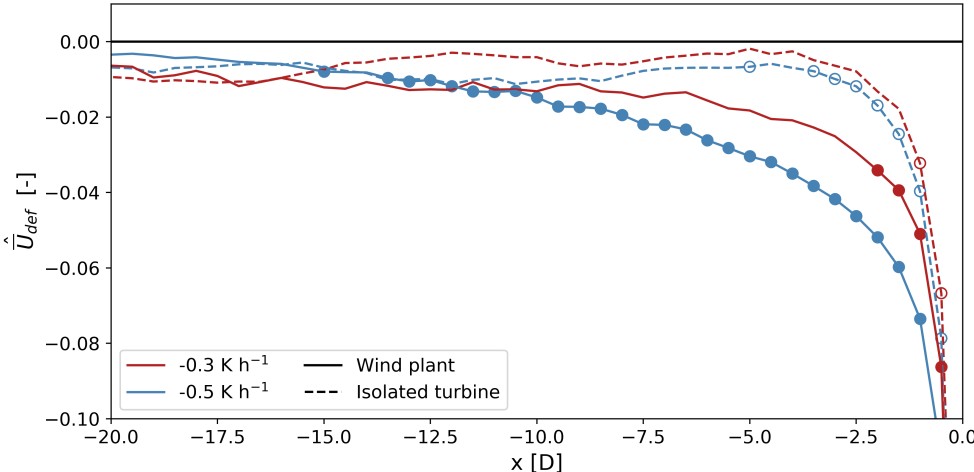

**Figure 5.** Normalized wind speed deficit upstream of an isolated turbine (dashed lines) and first row of turbines in wind plant (solid lines). The solid (empty) circles represent locations upstream in 0.5D increments where $\langle \overline{U} \rangle_{w/}$ and $\langle \overline{U} \rangle_{w/o}$ are statistically different ($\alpha = 0.05$) for each atmospheric condition for the wind plant (isolated turbine) case. The $x-$axis is re-scaled to locate $x = 0D$ at the first turbine row for the wind plant ($x = 6804m$) and at the rotor disk for the single turbine ($x = 5292m$).

Comparing the flow upstream of a single turbine and a wind plant indicates the induction zone extends much further upstream for a wind turbine array in both atmospheric conditions (solid and dashed lines of same color in Figure 5). The wind speed deficit 2.5D upstream of the rotor is $2.93\%$ ($0.79\%$) and $4.62\%$ ($1.18\%$) for the wind plant (isolated turbine) in the U12-C0.3 and U12-C0.5 cases, respectively. The wind speed deficit upstream of the wind plant remains at least 1% larger than for the single turbine up to $x = -9.5D$ for the U12-C0.5 case. Conversely, the wind speed deficit upstream only remains at least 1% larger for the wind plant compared to the single turbine up to $x = -6D$ for the U12-C0.3 case. Not only is the wind slowdown larger upstream of the wind plant compared to the single turbine, but also the difference in the observed deficit increases with increasing stability.

## 4   Boundary layer evolution

Differences in upstream blockage for both stability cases can be explained by evaluating the boundary layer evolution as the flow moves above and around the wind plant. We examine this evolution using the boundary layer height and turbulent momentum fluxes across the domain. The boundary layer height is defined as the local maximum in $d\theta/dz$ below the capping inversion. The turbulent fluxes are calculated from 5-minute averages of the velocity fields.

The vertical turbulent transport of zonal velocity restores momentum upstream of the plant for the U12-C0.3 case (Figure 6). Convergence of the $\overline{u'w'}$ momentum flux, especially upstream and downstream of the plant, dominates the turbulent momentum flux divergence budget, reintroducing zonal momentum upstream of the plant. As the flow moves through each turbine row,



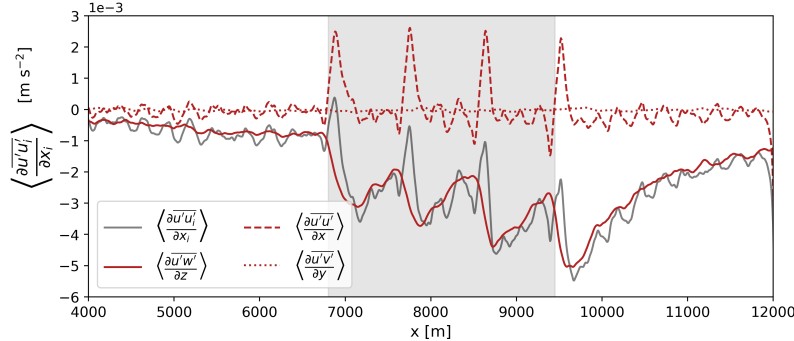

**Figure 6.** Turbulent flux divergence for the U12-C0.3 case, averaged in time and spatially across the wind plant ($y-$direction) and the turbine rotor layer ($z-$direction). The grey shaded region marks the location of the wind plant in the domain. The grey solid line represents the net turbulent flux divergence for the $u-$velocity.

divergence of the $\overline{u'u'}$ momentum flux increases, decelerating the flow. The sidewards divergence of momentum $\overline{u'v'}$ remains small over the entire domain, having negligible impact on the zonal velocity.

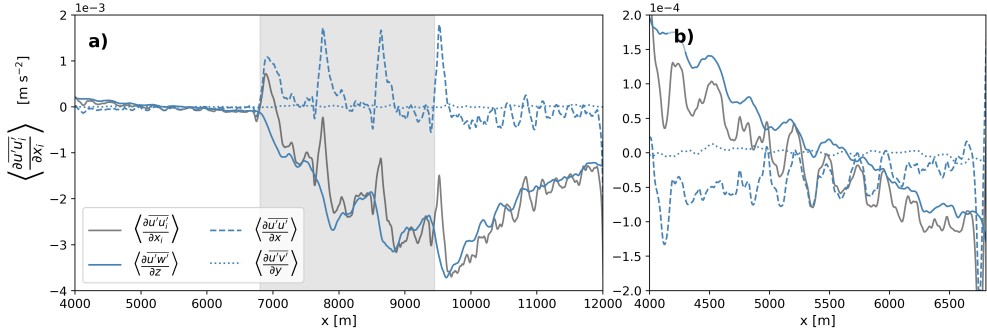

**Figure 7.** Turbulent flux divergence for the U12-C0.5 case over entire domain (a), averaged spatially across the wind plant ($y-$direction) and the turbine rotor layer ($z-$direction). The grey shaded region marks the location of the wind plant in the domain. The grey solid line represents the net turbulent flux divergence for the $u-$velocity. Panel (b) shows the same as (a), but zooming into the region upstream of the wind plant.

For the U12-C0.5 case, the turbulent transport of $u-$momentum is small upstream of the wind plant (Figure 7). Like in the U12-C0.3 case, the net divergence of turbulent momentum fluxes is still dominated by the convergence of the $\overline{u'w'}$ momen-
tum flux. However, these contributions become large and restore zonal momentum only after the first turbine row. Likewise, divergence of the $\overline{u'u'}$ momentum flux becomes important only as the flow moves through the turbine rotors. The sidewards divergence of momentum $\overline{u'v'}$ also stays small over the entire domain.

The wind plant deepens and cools the stable boundary layer by redistributing heat and momentum (Figure 8). The stable layer height increases more in the U12-C0.3 case compared to the U12-0.5 case as a result of stronger vertical turbulent motions





transporting heat and momentum. Enhanced mixing in the U12-C0.3 case homogenizes the potential temperature profile just above the turbine rotor layer, forcing the boundary layer height (max. in $d\theta/dz$) upwards. Conversely, mixing from turbulent vertical motions is hindered in the stronger stable layer (U12-C0.5 case), reducing the vertical displacement of the boundary layer top.

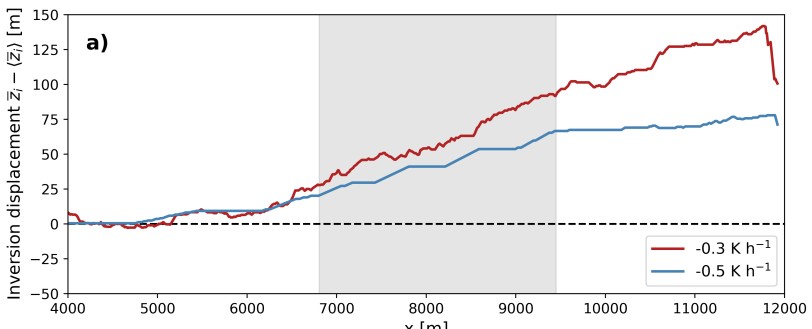 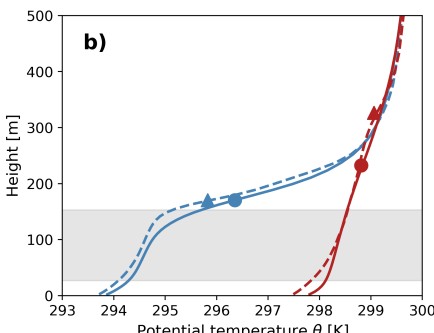

**Figure 8.** Inversion height displacement from unperturbed conditions (a), and potential temperature profile at multiple locations in the domain (b). The mean boundary layer height $\langle \overline{z}_i \rangle$ is calculated from the simulations without the GAD parameterization. The grey shaded region marks the location of the wind plant in the domain (a), and the turbine rotor layer (b). The solid (dashed) lines in (b) represent conditions upstream (downstream) of the wind plant. The circles and triangles in (b) represent the inversion height upstream and downstream of the wind plant, respectively.

## 5 Definition of freestream velocity

In Sect. 3 we quantified the upstream blockage effect using two sets of simulations, one with the GAD parameterization in the nested domain and one without. However, experimental studies do not often have the luxury of fully determining the unperturbed conditions upstream of a wind plant and so rely upon defining a freestream velocity that characterizes the flow (Bleeg et al., 2018; Schneemann et al., 2021; Segalini and Dahlberg, 2019). Local flow inhomogeneities and upstream blockage effects likely modify the freestream velocity, adding uncertainty to the extent and magnitude of the induction zone of a wind

plant. Here, we consider multiple methods of defining the freestream velocity within the simulation to mimic an experimental setup. We evaluate how each method changes the magnitude of wind slowdown captured upstream of the plant.

Our simulations display some variability in the streamwise "background" flow (Figure 9). Because this low-frequency signal occurs in both the simulations with and without the GAD parameterization, the turbines do not trigger this behavior in the flow. As the horizontal wind speed fluctuates throughout the domain, it displays lower winds at the inflow boundary of the nested

LES domain. As turbulent motions start developing at the inflow boundary, higher momentum is transported downwards in the boundary layer, increasing horizontal velocity at hub height. After $x = 4000$ m, turbulence starts decaying slowly through the rest of the domain (Figure 3), allowing winds aloft to develop an inertial oscillation that slightly increases the horizontal





velocity at hub-height. Though the magnitude of the hub-height horizontal wind speed fluctuations are small (max $\Delta U_{hh} \sim$ 0.25 m s$^{-1}$ for the U12-C0.5 case), these are of the same magnitude as the deficit caused by upstream blockage. Therefore, to

facilitate detection of the deficit, we remove this background flow from our time-averaged horizontal velocity fields:

$$\overline{U}_{noBkgd} = \overline{U}_{w/} - \left( \langle \overline{U} \rangle_{bkgd} - \langle \overline{U} \rangle_{\infty} \right), \tag{3}$$

where $\overline{U}_{w/}$ is the time-averaged velocity field in the simulations with the GAD, $\langle \overline{U} \rangle_{bkgd}$ is the time- and spatially-averaged (in the $y-$direction) velocity field in the simulations without the GAD, and $\langle \overline{U} \rangle_{\infty}$ is the mean velocity throughout the whole domain at a given height in the simulations without the GAD parameterization.

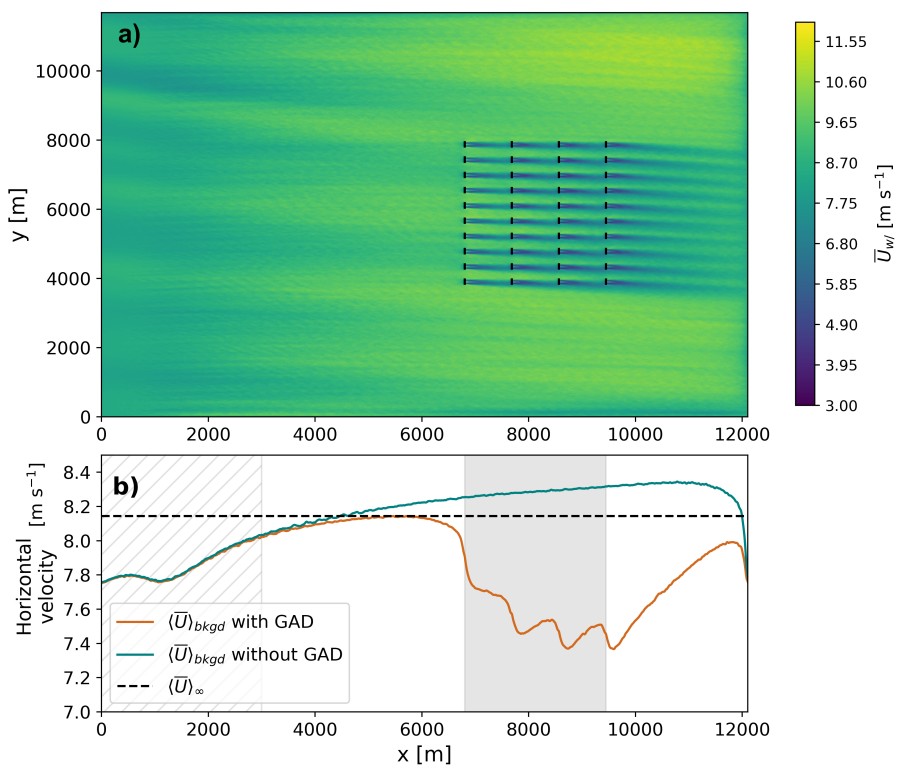

**Figure 9.** Time-averaged hub-height horizontal velocity contours for the U12-C0.5 simulation with the GAD parameterization (a). The solid black lines in (a) represent the wind turbines in the domain. Panel (b) illustrates the time- and spatially-averaged (in the $y-$direction) velocity field in the simulations with and without the GAD, and the mean velocity throughout the whole domain without the GAD parameterization. In (b), the hatched area represents the region where 3D turbulence is developing and is thus neglected. The grey shaded area represents the region covered by the wind turbines in the simulation with the GAD parameterization.

After removing the background flow from the velocity field, we calculate the freestream velocity upstream of the wind plant in multiple ways. We test five different approaches as shown in figure 10: 1) time-averaged hub-height wind speed measured at one point 10D upstream of the wind plant (referred to as "Single PM freestream"), such as would be available from a single



profiling lidar or meteorological tower; 2) time-averaged hub-height wind speed measured at three points 10D upstream of the wind plant (referred to as "Three PM freestream"); 3) time-averaged hub-height wind speed measured at six points 10D and

20D upstream of the wind plant (referred to as "Six PM freestream"); 4) time- and spatially-averaged hub-height wind speed measured over the area extending 1D to 20D upstream of the wind plant (referred to as "Area freestream"), such as would be available from a scanning lidar; and 5) time- and spatially-averaged hub-height wind speed measured over the whole turbulent domain of the no-turbine simulations (referred to as "True freestream"). For reference, wind tunnel experiments measured the freestream velocity using five point measurements at a constant distance upstream of the plant (Segalini and Dahlberg,

2019), similar to the setup shown in figure 10b. Field measurements by Bleeg et al. (2018), on the other hand, sampled inflow conditions using point measurements scattered upstream of the plant, similar to our arrangement in figure 10c. Scanning lidar measurements of the induction zone sampled the wind field from 40D to 5D upstream of the plant (Schneemann et al., 2021), resembling our layout in figure 10d.

Depending on how the freestream velocity is defined, the magnitude and spatial extent of the induction zone changes. For

the U12-C0.3 case, the difference between the freestream velocity estimated with a single point measurement and sampling the area upstream of the wind plant is 0.3 m s$^{-1}$, which is the same order of magnitude as the blockage effect we are trying to measure ($\sim$ 1%). For the U12-C0.5 case, the difference in the various definitions of the freestream velocity is smaller relative to the blockage effect but still present, differing by nearly 1% when comparing the velocity calculated using three point measurements and the area upstream.

Freestream velocity fluctuations exceed any detectable induction zone fluctuations (Figure 11). Similar to Sect. 3, we define the normalized velocity deficit using the single-valued freestream velocity as follows,

$$\hat{\overline{U}}_{def} = \frac{\langle \overline{U}_{noBkgd} \rangle - \langle \overline{U} \rangle_{\infty i}}{\langle \overline{U} \rangle_{\infty i}}, \tag{4}$$

where $\langle \overline{U}_{noBkgd} \rangle_{w/}$ is the time- and spatially-averaged velocity field for the simulations with the wind turbines that has had the background flow removed, and $\langle \overline{U} \rangle_{\infty i}$ is the freestream velocity estimated using the five different methodologies mentioned

above. We find different definitions of the freestream velocity augment or reduce the extent and magnitude of the induction zone. At $x = -2.5D$, the velocity deficit is 6.1%, 4.6%, 3.3%, and 3.4% (4.7%, 5.2%, 5.2%, and 4.3%) for the U12-C0.3 (U12-C0.5) case using "Single PM", "Three PM", "Six PM", and "Area" freestream velocities, respectively. At $x = -15D$ the differences remain just as large, but the magnitude of the velocity deficits becomes smaller at 4.2%, 2.8%, 1.4%, and 1.5% (0.7%, 1.3%, 1.2%, and 0.4%) for the U12-C0.3 (U12-C0.5) case using "Single PM", "Three PM", "Six PM", and "Area"

freestream velocities, respectively. As we increase the number of sampling locations to estimate the freestream velocity, the velocity deficit approaches the values obtained by using the "true" freestream of the flow. This behavior shows that an imprecise definition of the freestream velocity may produce errors of the same order of magnitude as the blockage effect. Further, a higher density of observations in a wider area is the most accurate approach to defining the freestream velocity, highlighting the utility of scanning lidars or radars for measuring blockage.





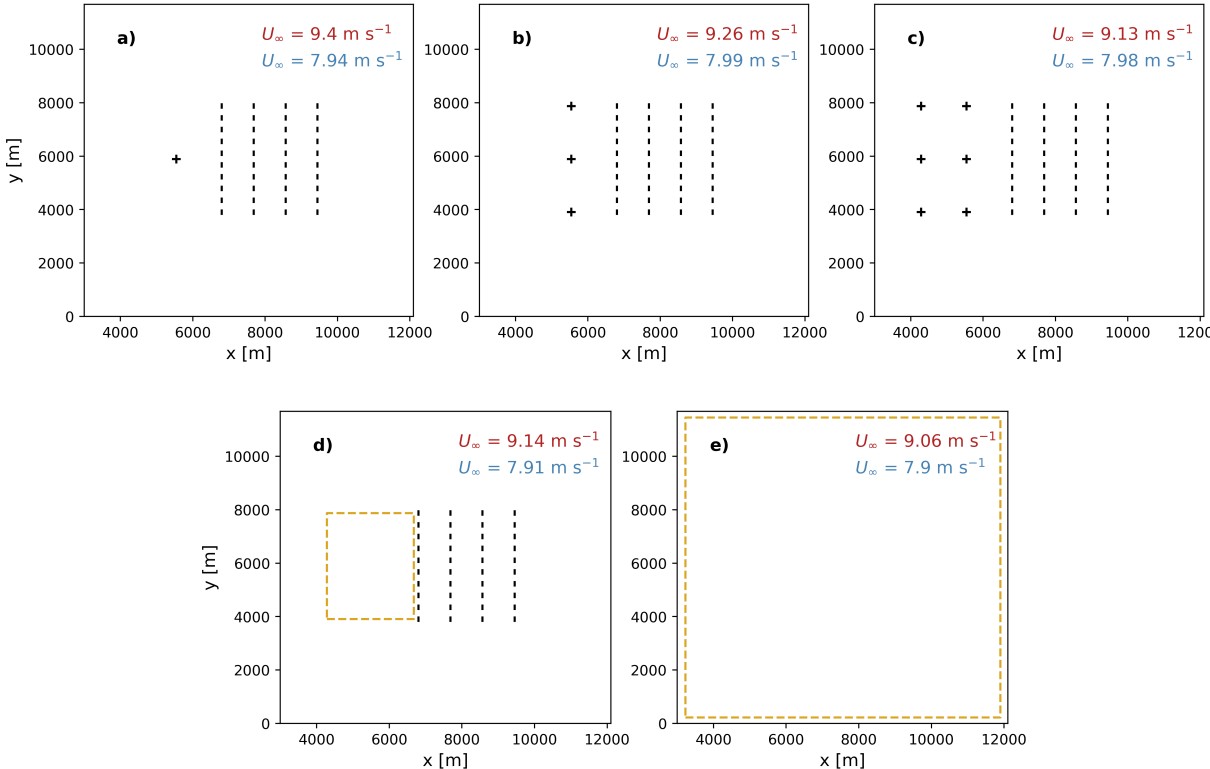

**Figure 10.** Schematic showing the relative location of the wind plant and the sampling locations for defining the freestream velocity of the flow. The freestream velocity in a), b), and c) is calculated using one, three, and six point measurements, respectively. In d) and e), the freestream velocity is calculated from areal measurements enclosed by the dashed yellow line. The solid vertical lines represent the individual wind turbines, as such panel e) represents the simulation with no turbines in the domain. The black crosses represent the locations for sampling the freestream velocity using point measurements. Freestream velocities in each panel are color coded for each stability condition: red (blue) text represents the U12-C0.3 (U12-C0.5) case.

## 6   Wind turbine power production

We now turn to analyze how wind turbine power production varies throughout the first row of the wind plant to understand how upstream blockage can undermine power production. We normalize turbine power for each turbine using the mean turbine power production for the first row of the plant as follows

$$\hat{\overline{P}}_i = \frac{\overline{P}_i}{\langle \overline{P} \rangle_{row=1}}, \tag{5}$$

where $\overline{P}_i$ is the time-averaged power production for turbine $i$, and $\langle \overline{P} \rangle_{row=1}$ is the mean turbine power for the first row of the wind plant. Further, we consider normalized power production for the turbines at the center (middle four turbines in row) and edges (two outer turbines on each side of row).





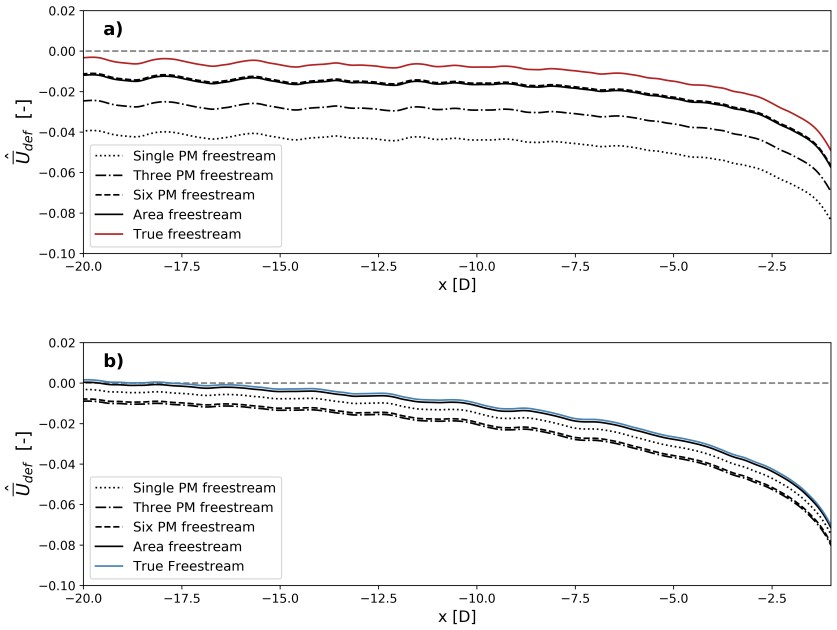

**Figure 11.** Normalized velocity deficit for the U12-C0.3 (a) and U12-C0.5 case (b) using the various definitions of freestream velocity. Results for the "True" Freestream velocities are color coded for each stability condition: red (blue) text represents the U12-C0.3 (U12-C0.5) case.

We evaluate the statistical significance of the power difference between center and edges turbines in the first row of the plant. Building on the assumption of a normally distributed velocity field, turbine power production is also considered normally distributed. As such, the $z$-statistic of the difference of the means is calculated as

$$z = \frac{\hat{\overline{P}}_c - \hat{\overline{P}}_e}{\sqrt{\frac{\sigma_c^2}{N_c^*} + \frac{\sigma_e^2}{N_e^*}}}, \tag{6}$$

where $\hat{\overline{P}}_c$ and $\hat{\overline{P}}_e$ are the normalized power for the center and edges turbines of the plant, respectively. The variance of normalized turbine power for the turbines at the center and edges of the wind farm are $\sigma_c$ and $\sigma_e$, respectively. The number of independent samples $N^*$ for each group of turbines is estimated using the total sample size $N$ and the lag-1 autocorrelation $\rho$.

Wind turbine power production varies throughout the first row of the wind plant (Figure 12). For the U12-C0.3 case, turbines at the center of each row of the wind plant generate more power than turbines at the edges. At the first row, normalized power is $\hat{\overline{P}}_c = 1.034$ and $\hat{\overline{P}}_e = 1.004$ for the center and edges turbines, respectively. However, the differences are not statistically significant ($\alpha = 0.05$) for any of the wind plant's rows. For the stronger stably stratified boundary layer case, the variability reverses such that the turbines at the center of the row consistently produce less power than turbines at the edges. At the first row, normalized power is $\hat{\overline{P}}_c = 0.982$ and $\hat{\overline{P}}_e = 1.024$ for the center and edges turbines, respectively. The difference in mean





turbine power production for the center and edges turbines is statistically significant for the first two rows of the wind plant, suggesting the blockage effect is stronger at the middle of the row compared to the edges.

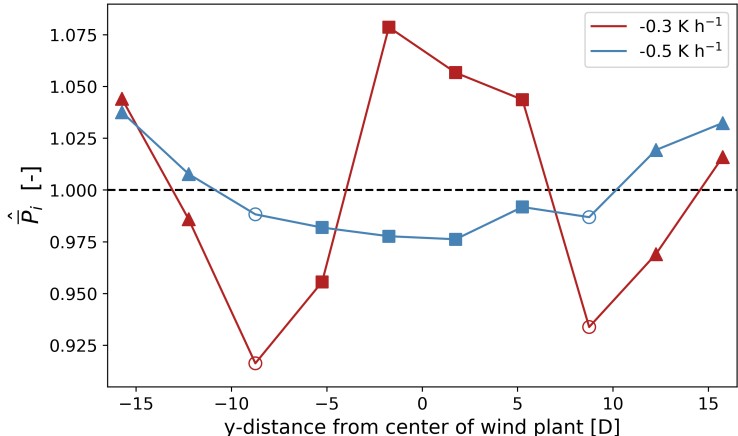

**Figure 12.** Normalized turbine power for the first row of the wind plant. Turbines at the center and edges of the plant are represented by squares and triangles, respectively. The turbines in between the center and the edges are represented by empty circles. The $x-$axis is re-scaled to locate $x = 0D$ at the center of the wind plant ($y = 5890m$).

## 7 Discussion and conclusions

Wind plant blockage undermines turbine power production in the first row of wind plants. As a result, energy production for operational wind plants is lower than expected (Ørsted, 2019; Bleeg et al., 2018). We use idealized LES in flat terrain to evaluate the upstream blockage effect for two distinct stable boundary layers, and demonstrate the influence of atmospheric stability on the induction zone using the boundary layer evolution across the domain. We also compare the induction zone of an isolated turbine to that of a large wind plant. To explore various methods for quantifying blockage from field measurements, we estimate the freestream velocity of the flow in different ways and evaluate how the velocity upstream diverges from this value. Furthermore, we evaluate turbine power production for the first row of the plant for each atmospheric condition.

The blockage effect from a wind turbine cluster is larger than for a turbine operating in isolation (Figure 5). To our knowledge, the only study that compares the flow upstream of a turbine with that of a wind plant for the same atmospheric conditions is that of Bleeg et al. (2018). Their RANS simulations show wind plants amplify the blockage effect compared to isolated turbines. They show that, while isolated turbines induce a wind speed slowdown around 1.5% 2D upstream, wind plants produce decelerations of around 4% at the same distance upstream (Bleeg et al., 2018). For the simulations herein, isolated turbines induce a wind speed slowdown of 1.3% (1.7%) 2D upstream, and wind plants produce decelerations of 3.4% (5.2%) at the same distance upstream, for the U12-C0.3 (U12-C0.5) case. Bleeg et al. (2018) also suggests isolated turbines do not influence the flow 7-10D upstream, but wind plants do. Our statistical analysis indicates that isolated turbines only significantly modify





the flow up to 5D upstream, while wind plants modify the flow up to 15D upstream. However, it should be noted that Bleeg
et al. (2018) do not provide a specific description of the stability cases used in their simulations, preventing a direct comparison.

Stronger stably stratified boundary layers amplify the upstream blockage effect in wind plants (Figure 5). A highly stratified
rotor layer hinders turbulent motions, especially in the vertical direction, reducing the downward momentum transport that
restores momentum to the flow and counteracts the cumulative blockage effect (Figures 6 and 7). Winds significantly slow
down far upstream compared to an unperturbed flow only in the boundary layer forced with a larger cooling rate at the surface
(U12-C0.5 case). Allaerts and Meyers (2018) show similar tendencies as a larger cooling rate at the surface largely increases
upstream blockage for their infinitely sized wind plant. Likewise, Wu and Porté-Agel (2017) show a larger lapse rate in the free
atmosphere above the neutral layer that contains the wind plant also amplifies the blockage effect. Experimental observations
also demonstrate that strong stable layers augment the blockage effect in wind plants (Schneemann et al., 2021). Though
studies agree that stability amplifies upstream blockage, there is discrepancy on the magnitude and extent of this effect.

The induction zone can extend up to ∼15D upstream and velocity deficits remains within the single-digit percentage range
(Figure 5). We do not find evidence of a far-reaching induction zone for the U12-C0.3 case. For this weaker stable layer, the
velocity field only displays statistically significant differences from an undisturbed flow up to 2.5D upstream of the wind plant.
Both Bleeg et al. (2018) and Schneemann et al. (2021) found single-digit decelerations upstream of wind plants. The average
3.4% and 1.9% wind slowdowns 2D and 7-10D upstream, respectively, of multiple wind plants surrounded by meteorological
masts found by Bleeg et al. (2018) shows good agreement with our findings. Similarly, Schneemann et al. (2021) found wind
speed is reduced by ∼4% between 30D and 5D upstream of an offshore wind plant. Though this deficit is larger compared
to our results, their uncertainty is in the order of ∼2%. Furthermore, Schneemann et al. (2021) combine a wide range of
stable cases, defined by the Obukhov length, in their findings. Differences in atmospheric static stability, wind plant layout and
method for measuring blockage may explain the subtle wind speed deficit discrepancies between our simulations and these
experimental results. Nonetheless, the physical mechanism driving upstream blockage in (Bleeg et al., 2018) and (Schneemann
et al., 2021) appears to be the same as in our study.

There is an order of magnitude difference in the extent and magnitude of the wind speed deficits upstream of wind plants
among LES studies, suggesting the driving mechanism for blockage changes. Wu and Porté-Agel (2017) show observable
wind speed decelerations up to 7km (88D) upstream, with deficits close to 10% 12.5D upstream of their finite-size wind plant.
However, their wind plant is nearly five times larger than the wind plant considered here, and their wind turbines are embedded
in a neutral boundary layer with stable stratification above rather in a stable boundary layer. Similarly, Allaerts and Meyers
(2018) find hub-height wind speed decelerations (∼10%) that extend upstream all the way to the inflow boundary of their
LES domain (7km or 70D). The most significant difference between our simulations is that they consider an infinitely large
wind plant in an incompressible flow, thus the streamwise flow slowdown is entirely transformed into vertical motions. Both
Wu and Porté-Agel (2017) and Allaerts and Meyers (2018) attribute these large (∼10%) flow decelerations to gravity waves
propagating upstream in their domains. A spectral analysis on the vertical and horizontal velocity at multiple locations in
our simulations shows no statistically significant evidence of waves moving through our domain. Thus, our simulations only



consider the effect of stability in the cumulative blockage effect of the turbines in the flow, rather than the effect of stability in triggering gravity waves upstream of wind plants.

As static stability in the surface layer increases, the turbulent vertical motions that would otherwise readily form are hindered by the stratified flow, restricting the boundary layer growth throughout the domain. Larger turbulent vertical motions in the weakly stratified flow erode the boundary layer top and deepen the stable layer more compared to the strongly stratified flow (Figure 8). Wu and Porté-Agel (2017) report similar results for their neutrally-stratified boundary layers even though the boundary layer height in their study extends far above the turbine rotor layer (above 500 m). Conversely, Allaerts and Meyers

(2017, 2018, 2019) show the maximum displacement of the boundary layer increases with stronger stability in their stable simulations, and with a lower and stronger capping inversion in their conventionally-neutral simulations. However, Allaerts and Meyers (2017, 2018, 2019) suggest gravity waves play an important role in their simulations, possibly indicating that the larger displacement in $z_i$ might result from more intense gravity wave activity in their stronger stable boundary layers.

     The inversion that characterizes the top of the stable boundary layer is forced upwards just before the upstream edge of

the wind plant and the maximum displacement occurs at the exit region (Figure 8). At the entrance of the wind plant, the mean flow is diverted upwards by the presence of the turbines. Further downstream, turbines' wakes enhance turbulent mixing, increasing the rate at which the boundary layer grows. Wu and Porté-Agel (2017) show matching tendencies for their weak free-atmosphere stratification case. However, the largest displacement occurs at the entrance of the wind plant for their strong free-atmosphere stratification simulation (Wu and Porté-Agel, 2017). Both stable cases in Allaerts and Meyers (2018) simulations

show the maximum displacement occurs at the wind plant's entrance region. The most likely explanation for this discrepancy is the upstream propagation of gravity waves. Every case that reports the maximum displacement of the boundary layer in the entrance region of the wind plant also attributes the large winds slowdowns to upstream propagating gravity waves (Wu and Porté-Agel, 2017; Allaerts and Meyers, 2018). The adverse pressure gradient that they show forms upstream of the wind plant diverts the flow vertically up in the entrance region, forcing the inversion height upwards. Conversely, a favorable pressure

gradient forms at the wind plant exit region, accelerating the winds and, due to incompressibility, instigating downward vertical motions that carry the boundary layer downwards. Our simulations, as well as the weak free-atmosphere stratification case from Wu and Porté-Agel (2017), on the other hand, do not evidence gravity waves.

     Upstream blockage has become a research priority because of effects on power production. Turbines at the center of the first row sometimes generate less power than those at the edges (Figure 12). A wind tunnel experiment by Segalini and Dahlberg

(2019) discovered similar results, where turbines at the center of a long wind plant experience winds less than 1% slower than turbines at the edges. For our simulations, mean power for the center turbines is smaller and statistically different from power generated by the turbines at the edges, but only for the stronger stability case (U12-C0.5). The weaker stability case displays a more inhomogeneous flow upstream as suggested by the variability in the freestream velocity (Figure 10). For the U12-C0.3 case, wind speed fluctuations in the cross-stream direction upstream of the first row of the plant surpass the wind

speed deceleration caused by wind plant blockage, resulting in the center turbines overperforming compared to turbines at the edges of the plant.



Choice of the freestream velocity can result in errors that are of the same magnitude as the estimated velocity deficit upstream of the wind plant (Figure 11). Bleeg et al. (2018) also uncovered similar errors when estimating the velocity deficit using meteorological masts in the vicinity and far away from operational wind plants. The magnitude of the wind speed slowdowns
captured by the met-masts close to the wind plant is sensitive to the choice of reference wind speed far away from the plant (Bleeg et al., 2018). They find up to 5% differences in the wind speed slowdown 2D upstream of a wind plant when using different reference conditions for the undisturbed airflow. Comparably, we find nearly 3% differences in the velocity deficit upstream induced by the wind plant for the U12-C0.3 case. This highlights how a freestream velocity that is not representative of the entire flow upstream, but rather affected by local flow inhomogeneities may obscure the actual induction zone of a wind
plant. Planning of field experiments that seek to quantify blockage, such as AWAKEN (Moriarty et al., 2020), should consider sampling an ample area upstream of the plant to properly define unperturbed flow conditions and distinguish the induction zone from inhomogeneities in the flow.

Thus, atmospheric static stability modifies upstream wind plant blockage by hindering turbulent motions, especially in the vertical direction, that replenish momentum in the flow. We find hub-height winds significantly decelerate up to 15D upstream
of the wind plant in a strongly stratified boundary layer. For weakly stratified flow, winds deceleration is only statistically significant up to 2.5D upstream of the wind plant. When estimating the induction zone of a wind plant using a freestream velocity, we find local flow inhomogeneities can produce freestream velocity variations that may exceed the velocity deficits upstream of the wind plant. The observed wind deceleration 2.5D upstream of the plant changes by nearly 3% (between 3.3% and 6.1%) for the weakly stratified boundary layer when using different methods for estimating the freestream velocity.

It is important to highlight that our simulations are idealized and in flat terrain. The role of terrain and vegetation should also be considered when evaluating wind plant blockage in future studies. Terrain forcing and differences in surface roughness modify mixing in the surface layer, likely altering the induction zone of wind plants. Furthermore, increased and inhomogeneous mixing in the surface layer add uncertainty to the definition of the freestream velocity in the flow. A higher degree of inhomogeneity in the inflow to the wind plant might obscure the existence of the induction zone when measured in field
experiments. Another important area of future research is the sensitivity of the blockage effect to wind speed, which is not explored herein. For example, at wind speeds above rated, turbines operate at lower thrust coefficients, reducing wake effects (Rhodes and Lundquist, 2013). It is likely that winds above the rated speed will alter the extent and magnitude of the wind plant's induction zone as well.

*Code and data availability.* The WRF model v4.1.5 used herein is available at: https://github.com/miguel-sg-2/WRF_versions.git. The namelist.input
and turbine locations files are available for download at DOI:10.5281/zenodo.4708020.

*Competing interests.* The authors declare that they have no conflict of interest.



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
