# Peer review of "Quantifying wind plant blockage under stable atmospheric conditions"

_Wind Energy Science, 2021_

## Referee Comment (RC2)

**General Comments**

Description of paper
The article 'Quantifying wind plant blockage under stable atmospheric conditions' by Gomez et al. draws conclusions about the magnitude and the sources of an observed velocity reduction upwind (blockage) of an idealized wind farm in two LES wind farm simulations. The main message of the article is that the blockage is higher in a stronger stratified atmospheric boundary layer and that the reason for this is the lack of vertical turbulent momentum transport. The authors further compare different virtual measurement setups to measure the wind speed upwind of the farm and analyse how a signal of blockage can be recognized in the production data. I see this work in general as a interesting addition to the current scientific discussion, but I see a couple of points that need to be addressed before I can recommend the full publication of the work.

**Mayor Comments**

1. Introduction: L. 26 - 35 - Definition of blockage, numbers

   I have an issue with the introduction of blockage in wind farms. I would say it is not proven that the upstream wind speed necessarily decreases more when turbines are combined in a wind farm. Furthermore I don't know any credible scientific publication that can relate the observed overpredictions of energy production of wind farms to blockage. The announcement of Orsted does not serve as a credible and sufficient reference. The numbers related to wake deficits (10 %) and blockage (1%) are not explained what they relate to. (wind speed? energy production?) If no reference is found here, I would suggest to rather talk about different orders of magnitude.

2. Duration of averaging and influence on results and conclusions

   The averaging period of 45 min appears quite small. If no longer averaging is possible, the limits of this restriction should be discussed throughout the paper. The paper draws conclusion on the significance of the results based also on the number of samples. For higher turbulence as in the lower stratification case, the significance is automatically lower. Thus, any conclusions about the significance of velocity differences, e.g. Figure 5, should point out that the signifigance criteria is strongly dependent on the length of measurement period. All conclusions about the significance of the results (wind speed deficit, power measurements) should relate to the selected sampling frequency (0.1 Hz) and the measurement period (45 min). A proper way to make the results between the two boundary layers more comparable is to scale the period of measurement to the turbulence level of the flow.

3. Discussion of measurement strategies (chapter 5)

   I have a hard time grasping the meaning of and the approach in this chapter. I understand the conclusion is that more measurement points (and thus more samples) reduce the uncertainty, which is I would say common sense. So, in this case it would be more interesting to look at the combination of different sampling frequencies and locations. Also, I don't understand why uncertainty is not displayed to evaluate the measurement setups, but rather a bias. Furthermore I don't think the averaging setups are even supposed to result into the same free stream velocity, as it can be clearly seen in Figures 3 and 9 that the flow is highly inhomogeneous

in x. In consequence my suggestion would be to either remove the chapter or put a lot more effort in working out the implications of the different measurement setups.

4. Conclusion on difference between the two simulations derived from flux divergence

I suggest to add the flux divergences from the simulations without any wind farms. As the flow does not appear to be stationary along x, I would assume that there is already divergence even without any wind farm. Like this I am still a bit skeptic to accept the difference in the vertical momentum flux to be the sole reason for the difference in upstream wind speed deficit. Also, what about the mean momentum fluxes?

**More Comments**

| | |
|---|---|
| L 1 | It's not true that only the first row of the plant is influenced |
| L 95 | *with* a *smaller time step* |
| Table 1 | should also have the height of the two domains |
| Figure 3 | The graph looks like a much longer domain would be necessary to derive at a quasi-stationary region along x. Were any sensitivity studies done for the choice of the simulation domain? |
| Figure 5 | Why are the lines not converging to zero at 20 D? |
| Figure 8 b | From Figure 8 a I would assume that the difference between the inversion height upstream and downstream should be a lot higher in the strongly stratified case than displayed here. |
| L 289 | See comment for L1 |
| Discussion & Conclusions | For me the chapter is too long and hard to read. I suggest to restructure the chapter. The part of the non-existent gravity waves for example could be written much shorter and more concise. |

---

## Author Comment (AC1)

**Anonymous Reviewer 1:**

Dear Anonymous Reviewer,

We highly appreciate your thoughtful feedback. It helped us to improve the manuscript and strengthen our findings. You highlighted some crucial points that were overlooked in our initial manuscript.

All reviewer comments appear in grey below, while authors' responses appear in blue text. Line numbers referenced in the authors' responses refer to the revised document. Figures included in the manuscript are labeled in italic and using numbers (e.g. *Figure 7* ), while figures that only appear in the response to reviewer comments are labeled in smaller font and using roman numerals (e.g. Figure iv).

**General Comments**

The authors present a study on wind-plant blockage in stable atmospheric conditions. Although this work is very timely, I'm not convinced that it is a useful addition to the existing literature. As further detailed below, there seem to be some very serious issues with the set-up, and a number of the results. I'm not sure whether this can be easily fixed – this could require rethinking the whole simulation setup.

**Major Comments**

1. The authors discuss an inertial oscillation that they later 'subtract' from their simulations (affecting the streamwise evolution of the flow). This seems a rude fix, and a good set-up should simply avoid this type of issue. Also, inertial oscillations occur in time (at a pretty low frequency) – how can they affect streamwise flow over such a relatively short fetch? Unfortunately, in combination with some of the strange results reported in the paper (cf. below), this issue raises some serious doubts on the correctness of the methodology.

Thank you for pointing out this imprecision in the text. We removed "inertial oscillation" from text because it does not accurately describe the mechanism behind the streamwise evolution of the flow. Furthermore, a streamwise evolution of the flow is not uncommon in limited area domain simulations that apply inflow turbulence generation techniques (Muñoz-Esparza et al., 2014, 2015; Muñoz-Esparza and Kosović, 2018). We now provide a description of the physical mechanism in the manuscript as follows:

L241 – L246: *"Our simulations also display some variability in the streamwise "background" flow (Figure 9b). The streamwise variability in the hub-height horizontal velocity results from turbulence development throughout the domain* (e.g. Muñoz-Esparza and Kosović, 2018). *As turbulent motions develop throughout the domain, higher momentum is transported downwards across the rotor layer, increasing the horizontal velocity at hub height. This downward transport of momentum and turbulence kinetic energy is often observed in stable boundary layers with low-level jets (Karipot et al., 2008; Banta et al., 2002; Mahrt and Vickers, 2002; Conangla and Cuxart, 2006; Wang et al., 2007)."*

We also analyzed in more detail the relationship between turbulence and horizontal velocity, and explored other approaches to removing the background flow in our simulations:

**Relation between turbulence and horizontal wind speed:**
To confirm this relationship between turbulence and horizontal wind speed, we perform a principal component analysis on the time-averaged horizontal velocity, and TKE across the domain for the simulations without the GAD parameterization. We evaluate the spatial pattern of each variable in the *x*- and *z*-directions for the U12-C0.3 simulation, which displays the largest streamwise variations in horizontal velocity.

The dominant spatial pattern for the evolution of TKE in the streamwise direction matches a dominant spatial pattern for the evolution of the horizontal velocity. The correlation in the empirical orthogonal functions of hub-height horizontal wind speed and TKE along the x-direction is 0.75 for the U12-C0.3 simulation. The empirical orthogonal functions for TKE and horizontal wind speeds at $y = 5200\ m$ (center of domain in y-direction) that explain 24% and 18% of variance, respectively, for the U12-C0.3 simulation are shown below.

[Figure]

Figure i: Empirical orthogonal functions for turbulence kinetic energy (top) and horizontal wind speed (bottom) at the center of the domain in the y-direction.

The empirical orthogonal functions demonstrate a downward vertical transport of turbulence kinetic energy and momentum, which result in an increase in hub-height wind speeds. It is common for stable boundary layers over land to display a downward transport of momentum in the presence of a low-level jet (Karipot et al., 2008; Banta et al., 2002; Mahrt and Vickers, 2002; Conangla and Cuxart, 2006; Wang et al., 2007), as we indeed have in both of our experimental setups.

**Background flow:**
We agree that subtracting the mean flow evolution from the simulations is not the most elegant approach. However, this approach is presented in order to prove our point, which is that quantifying blockage using field measurements can result in significant errors due to cross-stream flow inhomogeneities. Further, we tested different ways of removing the streamwise evolution of the flow, while retaining the cross-stream fluctuations that add uncertainty to the definition of the freestream velocity, and every other approach introduced more problems. We designed multiple Butterworth filters to remove the background flow of the simulation without the GAD parameterization. Then, we evaluated the filter in the simulations with the GAD parameterization to verify the signal from the turbines was not significantly affected. We tested multiple cut-off wave numbers for the high pass filter. The variance of the time-averaged streamwise velocity was used to evaluate the performance of the filter in removing the background flow. The figure below shows how the background flow is removed by using different cutoff wavenumbers.

[Figure]

Figure ii: Background flow for the simulation without turbines resulting from using different cut-off wavenumbers for the Butterworth filter.

The variance of the time-averaged horizontal velocity is significantly reduced using a second order high pass Butterworth filter with a cut-off wavenumber of 0.0002 m$^{-1}$ for the U12-C0.3 and U12-C0.5 simulations (Figure below).

[Figure]

Figure iii: Spatial variance of the time-averaged horizontal velocity for each atmospheric stability case as a function of the cut-off wavenumber.

Applying the 2$^{nd}$ order Butterworth filter to the simulation with the GAD parameterization shows minimal power is extracted at the scales of the same order as the streamwise turbine spacing ($k > 10^{-3}$ m$^{-1}$), while the background flow is still removed from the velocity field. However, the filter removes power from wavenumbers of the same order as the wind plant length ($k \sim 3 \times 10^{-3}$ m$^{-1}$), possibly affecting wake and blockage evolution throughout the domain.

[Figure]

Figure iv: Energy spectrum for the filtered and unfiltered velocities in the simulation with and without the GAD. The cut-off wavenumber is 0.0002 m$^{-1}$.

Filtering the background flow at hub-height removes the cross-stream inhomogeneities in the flow, and alters the evolution of blockage and wakes throughout the domain. The filter eliminates the differences in wind plant

blockage found in previous sections between both boundary layer simulations. Furthermore, the filtered velocity field displays an unphysical acceleration of hub-height winds in the wakes of the turbines.

This result demonstrates the filter is removing power from scales influenced by the wind plant, and thus this method is not adequate for removing the background flow from the simulations. Conversely, subtracting the background flow from the simulations does not produce unphysical results and is consistent with our argument, which is that quantifying blockage using field measurements can result in significant errors due to cross-stream flow inhomogeneities.

[Figure]

Figure v: Plan view of the filtered and unfiltered velocity field at hub height for the U12-C0.3 simulation. Note that the turbine wakes in the filtered velocity field display higher velocities than the inflow to the wind plant.

[Figure]

Figure vi: Time and spatial (in y-direction) average of the filtered hub-height horizontal velocity deficit for each stability case.

2. There are serious spanwise fluctuations in the inflow velocity (see, e.g., Figure 9). State of the art LES simply does not have this kind of problem. Also there is no real analysis on the cause of this issue. Presumably this comes from the parent domain, but not much analysis is performed (are these streaks existing in the parent domain as well, is this a result of the coupling methods, …).

We modified the manuscript to include a description of these elongated structures, which indeed propagate from the parent domain (see figure below). These elongated structures of alternating high- and low- wind speeds are commonly observed in neutral and stable boundary layer simulations with several types of LES (e.g. Peña et al., 2021; Mirocha et al., 2018; Moeng et al., 2007; Saiki et al., 2000; Moeng and Sullivan, 1994). The manuscript is modified to include the following:

L238 – L241: *"The inflow velocity for the nested domain exhibits cross-stream fluctuations that erode downstream of the inflow boundary. These elongated structures propagate from the parent LES domain and add variability to the flow upstream of the wind plant (Figure 9a). These structures are commonly found when simulating stable and neutral boundary layers (Peña et al.,2021; Mirocha et al., 2018; Moeng et al., 2007; Saiki et al., 2000; Moeng and Sullivan, 1994)."*

[Figure]

Figure vii: Instantaneous, hub-height horizontal velocity field for the parent domain (Δx = 70m) of the simulations for each stability case. The grey dotted line represents the location of the nested domain within the parent domain.

3. The authors use a 500m Rayleigh damping layer to avoid reflection of gravity waves. However, nonreflecting damping layers are tricky. I would expect that the layer should be at least one, possibly better two dominant vertical wavelengths. What is the vertical wavelength that can be expected based on wind-farm length and Brunt–Väisälä frequency? Can you report the level of reflection in your simulation – this can, e.g., be simply estimated using the method of Taylor and Sarkar (JFM 2007).

Thank you for raising this interesting point. We estimate the vertical wavelength of gravity waves using:

$$\lambda_z = \frac{2\pi \langle U \rangle_z}{\langle N \rangle_z}$$

Since we have multiple temperature stratifications in our domain (see figure below), the vertical wavelength of gravity waves for each vertical temperature stratification is as follows:
- $\lambda_z$ in the underline{surface} layer is 896 m (1964 m) for the -0.5 K/h (-0.3 K/h) simulation.
- $\lambda_z$ in the underline{residual} layer is 1867 m (2042 m) for the -0.5 K/h (-0.3 K/h) simulation.
- $\lambda_z$ in the underline{troposphere} is 1032 m (1021 m) for the -0.5 K/h (-0.3 K/h) simulation.

Though $\lambda_z$ is about 2 km in the residual layer, the vertical wavelength is 1000 m in the troposphere, where the damping layer is located in our simulations. Klemp and Lilly (1978) found the depth of the damping layer should be of the order of one vertical wavelength, which is not far off in either of our simulations.

[Figure]

Figure viii: Streamwise evolution of the Brünt-Väisälä frequency for each stability case for the nested domain at y = 5200m. Note the white contours represent values outside the colorbar, resulting from the potential temperature perturbations (CPM) at the inflow boundary of the domain.

4. Figure 5: the authors claim that upstream of the symbols marked on the figure, there is no significant measurable blockage effect. Why is it then that all simulations still provide a negative deficit far upstream – I would expect, statistically speaking, some of them to be positive. The chance at four heads is only about 6%.

We appreciate this comment and we added clarification in the manuscript. Our simulations show there is indeed an effect, however, the effect is not statistically significant (95% confidence level). We define the induction zone using a statistical analysis on the velocity fields. Therefore, the induction zone does not extend that far upstream because our analysis demonstrates this small deviation is not statistically significant.

The figure below shows the confidence intervals on the normalized velocity deficit from the strong stably stratified simulation. This shows the mean velocity deficit is slightly less than zero at 20D, but the confidence intervals are not below zero. Therefore, the induction zone as we define it does not extend beyond 20D. Confidence intervals are not included in Figure 6 in the manuscript because we want to aggregate the results from the different cases (i.e. different stability cases, and single turbine and wind plant) into one same plot, while showing the statistical significance of our results.

[Figure]

Figure ix: Normalized velocity deficit for the U12-C0.5 case. The error bars represent the 95% confidence intervals.

The manuscript is modified as follows:
L168-169: "*We define the induction zone with the statistical significance of the velocity deficit upstream of the first row of turbines in the plant.*"

We also modify the way in which we describe statistical significance. We now mention confidence intervals rather than the z-statistic as follows:

L171-176: "*The 95% confidence interval ($\alpha$= 0.05) of the difference of means, $\langle \overline{U}_w \rangle_y - \langle \overline{U}_{wo} \rangle_y$, at each x–distance upstream is calculated as*

$$CI = \pm z_{\alpha/2} \sqrt{\frac{\sigma_w^2}{N_w^*} + \frac{\sigma_{wo}^2}{N_{wo}^*}}$$

*where $\sigma_w$ and $\sigma_{wo}$ are the variance of the velocity field in the simulations with and without the turbines, respectively. The z–statistic $z_{\alpha/2}$ for the 95% confidence level is 1.96. The number of independent samples at each distance upstream, $N^* = N \frac{1-\rho}{1+\rho}$, is estimated using the total sample size, N, and the lag-1 autocorrelation $\rho = \overline{U'(t)U'(t + \Delta t)}/\overline{U'(t)^2}$, (Wilks, 2019).*"

L180-181: "*From this point onward, a statistically significant velocity deficit is such that its 95% confidence interval does not contain the value of 0 $\left( \overline{U}_{def} \pm CI \notin 0 \right)$.*"

For clarity, we also included:

L186-188: "*The wind plant modifies the flow in a statistically significant manner up to 15D upstream for U12-C0.5. Conversely, there is not enough statistical evidence that the induction zone extends further than 2D upstream for U12-C0.3.*"

L345-346: *"We do not find statistical evidence of a far-reaching induction zone for the U12-C0.3 case. For this weaker stable layer, the velocity deficit is only statistically significant up to 2.5D upstream of the wind plant."*

5. Figure 8. The inversion displacement keeps growing downstream of the farm. I would expect that it goes down again. You seem to define z_i based on max of dtheta/dz . If so, this measure would include possible turbulent mixing at the interface (thus overestimating displacement, which should be based on a streamline). However, more problematic is that there should not be any turbulent mixing at the interface in a stable boundary layer situation. Looking at your forcing methods, it seems that you force up to the capping inversion, so also in the residual layer, which should not have any turbulence. If correct, this does not make any sense!!

Thank you for the constructive feedback, it helped us describe better our simulation setup. We trigger turbulent motions up to the capping inversion at the inflow boundary. However, the $\langle \overline{w'w'} \rangle$ profiles below show turbulence in the residual layer has largely decayed after x = 4000 m. Furthermore, observations in regions with flat terrain demonstrate turbulence in the residual layer is not necessarily zero (e.g. Banta et al., 2006; Banta, 2008; Bonin et al., 2020). Ultimately, turbulence fluctuations will be removed by the model despite any boundary conditions forcing if the static stability is strong enough to suppress mechanical turbulence production. However, capping inversions often display some turbulence activity as a result from entrainment processes and the strong gradients that occur in the vicinity of this layer.

We modify section 2.3 in our manuscript as follows:

L147-154: *"We determine whether turbulence has propagated throughout the entire domain using the variance of the vertical velocity, which is calculated using 20 min time windows. Turbulence is close to steady 20 min after initializing the nested domain for U12-C0.3 (Figure 3a), and 30 min after initializing the nested domain for U12-C0.5 (Figure 3b), and results are discarded before these times. Furthermore, turbulence in the surface layer becomes quasi-stationary after x = 4000 m for both simulations (Figure 4), and we also discard results upstream of this location. Although we trigger turbulent motions up to the capping inversion, turbulence in the residual layer decays rapidly throughout the domain and becomes small after x = 4000 m. Note that minimal turbulence persists in the residual layer, as is sometimes observed in regions with flat terrain (Banta et al., 2006; Banta, 2008; Bonin et al., 2020).*

*Figure 3: Time evolution of the vertical velocity variance for the U12-C0.3 (a), and U12-C0.5 (b) simulation without the wind turbines. The profiles are averaged over a 0.8 km² region centered at the first row of the wind plant (x = 6804 m, y = 5890 m). The perturbations of the vertical velocity are calculated using a 20-min moving average.*

[Figure]

*Figure 4: Evolution of the vertical velocity variance averaged in the y-direction across the domain for the U12-C0.3 (a), and U12-C0.5 (b) simulations without the turbines. Vertical profiles are color coded for each x-location in the domain and plotted in 1000 m increments.*

[Figure]

Our definition of the inversion height does include turbulent mixing at the interface. However, this mixing is primarily from turbine-generated turbulence. Turbulence generated by the turbines enhances mixing across the stable boundary layer and smooths the potential temperature profile. In such a way, the inversion height should increase as turbine-generated turbulence decays downstream of the wind plant. Nevertheless, turbulent mixing is more restricted in the -0.5 K/h case compared to the -0.3 K/h case due to the strong temperature inversion at ~170m. We modified the text as follows:

L220-228: *"The vertical displacement of the flow upstream and over the wind plant deepens and warms the stable boundary layer (Figure8 a,b). The boundary layer starts growing upstream of the wind plant as a result of vertical advection of heat and momentum by the mean flow. Boundary layer growth in this region is practically the same for both stable boundary layers (Figure 8a). In contrast, the inversion height is displaced more in U12-C0.3 compared to U12-C0.5 after the first turbine row.*
*The inversion height evolution downstream of the first row of the wind plant is determined by turbulence. A deeper internal boundary layer characterized by enhanced turbulence levels forms in U12-C0.3 compared to U12-C0.5 over the wind plant (Figure 8c). Enhanced mixing in U12-C0.3 homogenizes the potential temperature profile just above the turbine rotor layer, forcing the boundary layer height (maximum in $d\theta/dz$) upwards. Conversely, mixing from turbulent vertical motions is hindered in the stronger stable layer (U12-C0.5 case), reducing the vertical displacement of the boundary layer top.*

*Figure 8: Inversion height displacement from unperturbed conditions (a), potential temperature profile at multiple locations in the domain (b), and depth of turbine-generated turbulent layer (c). The mean boundary layer height, $\langle \overline{z}_i \rangle$, is calculated from the simulations without the GAD parameterization. The grey shaded region marks the location of the wind plant in the domain (a,c), and the turbine rotor layer (b). The solid and dashed lines in (b) represent conditions upstream and downstream of the wind plant, respectively. The circles and triangles in (b) represent the inversion height upstream and downstream of the wind plant, respectively.*

[Figure]

We also tested other definitions of boundary layer height, including a streamline and the height of $1.05\overline{U}_g$, and we found similar results. The different definitions of the boundary layer height showed the boundary layer is displaced upwards more in U12-C0.3 compared to U12-C0.5, and the maximum displacement occurs at the exit of the wind plant. Therefore, we decided to retain our current definition.

6. Section 4, and the analysis related to Figure 6 and 7: I'm not sure what exactly the point is of this exercise (apart from the fact that it is possible). Also, turbulent flux divergence is not enough to study the momentum balance. Other terms that definitely seem relevant are the mean momentum transport (e.g. in the entrance region of the farm), which are not discussed here. Other terms (e.g. pressure forcing) are probably negligible, but this should be discussed.

We appreciate the motivation to better support our analysis, and we agree other terms are also very relevant. We analyze the following terms, and how they differ from the simulations without the GAD: $\overline{u}_i\frac{\partial \overline{u}}{\partial x_i}$, $\frac{1}{\rho}\frac{\partial \overline{p}}{\partial x}$, and $\frac{\partial \overline{u'u'_i}}{\partial x_i}$. The induction zone seems to be most impacted by $\overline{w}\frac{\partial \overline{u}}{\partial z}$. The dominant terms in the x-momentum equation are $\overline{u}\frac{\partial \overline{u}}{\partial x}$, $\overline{w}\frac{\partial \overline{u}}{\partial z}$, and $\frac{1}{\rho}\frac{\partial \overline{p}}{\partial x}$. The pressure gradient term $\left(\frac{1}{\rho}\frac{\partial \overline{p}}{\partial x}\right)$ is nearly identical for both simulations, thus the difference in the wind plant's induction zone is largely determined by the advection of x-momentum by the vertical velocity $\left(\overline{w}\frac{\partial \overline{u}}{\partial z}\right)$. The negative vertical transport of x-momentum across the rotor layer is larger for the -0.5 K h$^{-1}$ simulation compared to the -0.3 K h$^{-1}$ simulation due to stronger vertical shear of the horizontal velocity. This larger negative vertical transport of x-momentum in turn requires a larger positive streamwise transport of x-momentum for the -0.5 K h$^{-1}$ simulation compared to the -0.3 K h$^{-1}$ simulation. We modified the manuscript as follows:

L203-219: "*In evaluating the x-momentum equation, we assume the flow is steady. This is a fair assumption because the cooling rate at the surface, which drives unsteadiness in the flow, results in small changes over the 40-min time averaging period. Furthermore, we neglect the Coriolis force in this analysis because the domain size (12000 m) is small compared to the scales affected by the Earth's rotation L=U/f=$\mathcal{O}(10^5\ m)$. In such a way, we consider momentum advection by the mean flow, pressure divergence, and the divergence of turbulent momentum fluxes. The turbulent fluxes are calculated from 20-minute averages of the velocity fields.*

*Figure 7: Mean flow momentum advection (a), and pressure gradient (b) terms of the x-momentum equation averaged spatially across the wind plant (y-direction) and the turbine rotor layer (z-direction). The plots show the departure of the terms in the momentum equation from the flow without the GAD.*

[Figure]

*The induction zone of the wind plant is most affected by the vertical transport of zonal momentum across the rotor layer (Figure 7). The pressure gradient term remains nearly equal for the U12-C0.5 and U12-C0.3 simulations (Figure 7b), given that the drag exerted by the turbines on the flow is very similar for both cases. Conversely, the negative vertical transport of x-momentum across the rotor layer is larger for U12-C0.5 compared to U12-C0.3 due to stronger vertical shear of the horizontal velocity (solid line in Figure 7a). As the flow is forced to move above the wind plant, the vertical momentum transport is balanced by the streamwise momentum advection. The larger vertical momentum loss in U12-C0.5 compared to U12-C0.3 requires additional streamwise advection of x-momentum for the flow to remain steady (Figure 7a), producing more flow deceleration up to 10D upwind of the wind plant. Turbulence divergence plays a minor role in the region upwind of the wind plant (not shown). Though the turbulence divergence terms are larger for U12-C0.3 compared to U12-C0.5, these remain virtually unchanged for the simulation with and without the GAD, suggesting they do not contribute significantly to momentum replenishment upwind of the turbines."*

We no longer consider turbulent momentum fluxes to play a major role in the induction zone of the wind plant. The turbulent momentum flux divergence terms still act to replenish momentum in the U12-C0.3 simulation, however, these terms are one order of magnitude smaller than the mean flow momentum transport and pressure divergence terms. Furthermore, we compare these terms for the simulations with and without the GAD (solid and dashed lines below) and there is virtually no difference for both simulations. This implies turbulent momentum fluxes do not influence the induction zone of the wind plant.

[Figure]

Figure x: Turbulence momentum flux divergence terms for the U12-C0.3 (red) and U12-C0.5 (blue) simulations. The solid and dashed lines represent the fluxes for the simulation with and without the GAD, respectively.

We also modified the discussion section accordingly and removed the discussion on turbulent momentum fluxes:

L337-339: *"A highly stratified atmosphere hinders turbulent motions, increasing vertical shear of the horizontal velocity and thus modifying mean momentum advection across the rotor layer."*

7. Figure 11: as far as I understand, the figure discusses the effect of using a wrong upstream reference to define the normalized velocity deficit 2.5D upstream of the farm (a measure for the blockage). I would expect that using a reference that also is 2.5D upstream, should lead to a deficit that is zero. Any reference that is taken farther upstream should than lead to a larger deficit… Why is it then that using the true freestream velocity leads to the lowest deficit 2.5D upstream???

This comment was very helpful and helped us clarify this section. Though Figure 12 does show the effect of using a wrong upstream reference to quantify blockage, we do not use a reference that is 2.5D upstream. Rather we test different ways of defining the reference velocity (Figure 10). The one closest to the wind plant is 10D upstream as stated in L257-258. We modified how we reference each methodology as follows:

L256-263: *"After removing the background flow from the velocity field, we calculate the freestream velocity upstream of the wind plant in multiple ways. We test five different approaches as shown in Fig. 10: 1) time-averaged, hub-height wind speed measured at one point 10D upstream of the wind plant $\left(\overline{U}_{\infty_{1PM}}\right)$, such as would be available from a single profiling lidar or meteorological tower; 2) time-averaged, hub-height wind speed measured at three points 10D upstream of the wind plant $\left(\overline{U}_{\infty_{3PM}}\right)$; 3) time-averaged, hub-height wind speed measured at six points 10D and 20D upstream of the wind plant $\left(\overline{U}_{\infty_{6PM}}\right)$; 4) time- and spatially averaged hub-height wind speed measured over the area extending 1D to 20D upstream of the wind plant $\left(\overline{U}_{\infty_{A}}\right)$, such as would be available from a scanning lidar; and 5) time- and spatially averaged hub-height wind speed measured over the whole turbulent domain of the no-turbine simulations (referred to as "True freestream" $\overline{U}_{\infty_{T}}$).*"

Furthermore, the convention for $\overline{U}_{\infty_i}$ can be directly related to Figure 10, and Figure 12 now includes this convention as well.

*"Figure 10: Schematic showing the relative location of the wind plant and the sampling locations for defining the freestream velocity of the flow. The freestream velocity in (a), (b), and (c) is calculated using one-, three-, and six-point measurements (PM), respectively. In (d) and (e) the freestream velocity is calculated from areal measurements enclosed by the dashed yellow line. The solid vertical lines represent the individual wind turbines. As such panel (e) represents the simulation with no turbines in the domain. The black crosses represent the locations for sampling the freestream velocity using point measurements. Freestream velocities in each panel are color-coded for each stability condition: red (blue) text represents the U12-C0.3 (U12-C0.5) case.*

[Figure]

*Figure 12: Normalized velocity deficit for the (a) U12-C0.3 and (b) U12-C0.5 case using the various definitions of freestream velocity shown in Fig. 10. Results for the True freestream velocities are color-coded for each stability condition: red (blue) text represents the U12-C0.3 (U12-C0.5) case. Note that the colored lines in (a) and (b) are not the same as the corresponding lines in Figure 6 because here the freestream is single-valued, whereas in Figure 6 the freestream varies in the streamwise direction. Confidence intervals are not shown.*

[Figure]

None of these approaches necessarily lead to a deficit that is zero, because there are cross-stream fluctuations in the velocity field across the domain and the sampling locations are not directly in front of every turbine.

**Minor Comments**

- Abstract, first phrase: blockage does not just impact on the performance of the first row

Thank you for pointing out this imprecision, we modified Abstract and Discussion sections accordingly.

- page 4: a sketch of the nested domains would be useful

Though it is common practice to show how we nest our domains, we are using periodic boundary conditions for the parent domain making this unnecessary. Below is the relative location of the nested domain within the parent domain, but we did not include this in the paper because it does not add insight into the simulations.

[Figure]

Figure xi: Relative location of the wind plant and the nested domain within the parent domain.

- page 5: can you better justify the combination of surface cooling with 9h30min spin-up? This is an overland situation – most of the night has already been passed after 9+ hours. (I realize that often long spin-up is used to arrive at some sort of steady state in SBLs, but given that you are forcing with a mesoscale model, this feels unnecessary?)

Spin-up of the parent domain was evaluated using hub-height wind speed and wind direction, and the potential temperature profile. Spin-up of the parent domain was finalized when hub-height winds at the center of the

parent domain reached a quasi-steady state and the inversion strength reached a certain threshold for each case. Given that we were looking for a stronger stable layer using the -0.5 K h$^{-1}$ cooling rate, spin-up time for this simulation was longer to reach the desired potential temperature profile. However, it is not possible to achieve a completely steady state since there is a cooling rate at the surface that continually forces the stable surface layer.

[Figure]

Figure xii: Time evolution of the wind direction (top) and hub-height wind speed (bottom) from the initial conditions.

We modified the manuscript as follows:

L111-114: "*Spin-up of the parent domain is complete when winds at the center of the parent domain reach a quasi-steady state and the inversion strength was 2 and 5 K for U12-C0.3 and U12-C0.5, respectively. Steadiness of hub-height winds is evaluated using the wind direction and wind speed upstream of the wind plant location.*"

- page 5: what is geostrophic wind? What is pressure gradient – are you using a geostrophic balance and barotropic conditions?

Our simulations are not in geostrophic balance nor with barotropic conditions. By mentioning the geostrophic in the simulation setup, we are following a commonly terminology used in the atmospheric science community when referring to how winds are initialized in the simulation. We removed this from the manuscript to avoid this confusion.

- Figure 2: would be interesting to see the profiles op to the top of the domain (up to 2500m). Also, can you add the slopes 0.01 K/m and 0.001 K/m into Fig. 2a

We modified Figure 2 to include the profiles up to 2km and information about the initial conditions. There is not much information that can be gained from showing the potential temperature profiles up to the domain top or by showing the slopes of the potential temperature inversions. The potential temperature and wind speed profile above 1500 m remain unchanged for both stability cases. The inversion strength at the end of the simulation changes slightly with time due to mixing and discontinuities in the initial specified profile.

[Figure]

Figure xiii: Temporal evolution of the potential temperature (top panels) and horizontal wind speed (bottom panels) profiles for the parent domain in each simulation. The left and right columns illustrate the U12-C0.3 and U12-C0.5 stability conditions, respectively.

We modified Figure 2 of the manuscript to include the initial potential temperature and wind speed profiles for the simulation:

*"Figure 2: Potential temperature (a) and wind speed (b) profiles at x = 4500 m and y = 5840 m averaged in time for the simulations without the wind turbines. The turbine rotor layer corresponds to the grey, shaded region. The black, dotted lines represent the initial conditions for the parent domain.*

[Figure]

"

- line 147: please improve sentence

We modified this sentence as follows:

L147-148: "*We determine whether turbulence has propagated throughout the entire domain using the variance of the vertical velocity, which is calculated using 20 min time windows.*"

- Use of brackets for spatial averaging is ambiguous: sometimes it is averaging in y, sometimes in x and y (eg. in eq 3). Please improve notation throughout the paper for clarity

We appreciate this comment. We agree there is ambiguity in spatial averaging and the convention follows throughout the paper. We modified the entire manuscript to follow the convention outlined in L165-167:

L165-167: "*Note that in Eq. 1 and what follows, an overbar ( ‾ ) denotes time averaging, angled brackets ⟨ ⟩$_i$ denote spatial averaging along the i-direction, and a hat ( ^ ) denotes a normalized quantity.*"

- Figure 4: explain in caption that averaging is only in y-direction

The manuscript was modified accordingly.

- z-statistic and \alpha: not clear from the text how they are related. Also lag-1 autocorrelation: not defined, no reference. Please provide a decent statistical analysis. I would prefer 95% confidence intervals on the results in Figure 5 (let the reader appreciate what differences are significant or not). Also statistical analysis can be based on moving block bootstrapping, rather than handwaving arguments on possible gaussian distribution and a proxy for integral time scale (which is presumably what you are implicitly doing with the lag-1 autocorrelation)

Thank you for this helpful suggestion. The figures below show the velocity distribution at each grid cell upstream of the wind farm is very close to Gaussian (Skewness close to zero, and Kurtosis close to three). Thus, the assumption of Gaussianity is appropriate. We modified the description of the statistical analysis to include confidence intervals rather than the z-statistic because this may be more understandable for readers. Also, we provided a brief description of the meaning of statistical significance.

[Figure]

Figure xiv: Streamwise evolution of Skewness (top) and Kurtosis (bottom) upstream of the wind plant for each atmospheric condition and the simulations with and without the wind plant. The black dotted lines in each plot illustrate the values for a Gaussian distribution.

The manuscript is modified as follows:

L169-171: *"The velocity distribution at each grid point upwind of the wind plant is close to Gaussian. In average, the Skewness and Kurtosis of the velocity distributions upstream of the wind plant is -0.05 and 2.1 (0.15 and 2.23) for the U12-C0.3 (U12-C0.5) simulation, respectively."*

L171-176: *"The 95% confidence interval (α= 0.05) of the difference of means, $\langle \overline{U}_w \rangle_y - \langle \overline{U}_{wo} \rangle_y$, at each x–distance upstream is calculated as*

$$CI = \pm z_{\alpha/2} \sqrt{\frac{\sigma_w^2}{N_w^*} + \frac{\sigma_{wo}^2}{N_{wo}^*}}$$

*where $\sigma_w$ and $\sigma_{wo}$ are the variance of the velocity field in the simulations with and without the turbines, respectively. The z–statistic $z_{\alpha/2}$ for the 95% confidence level is 1.96. The number of independent samples at each distance upstream, $N^* = N \frac{1-\rho}{1+\rho}$, is estimated using the total sample size, N, and the lag-1 autocorrelation $\rho = \overline{U'(t)U'(t+\Delta t)}/\overline{U'(t)^2}$, (Wilks, 2019)."*

L180-181: *"From this point onward, a statistically significant velocity deficit is such that its 95% confidence interval does not contain the value of 0 $\left( \overline{U}_{def} \pm CI \notin 0 \right)$."*

- line 172: not a proper sentence

We modified the sentence as follows:

L182-183: *"The normalized velocity deficit along the x-direction for the wind plant and single turbine demonstrates that stronger stable stratification amplifies upstream blockage."*

- line 192: "The turbulent fluxes are calculated from 5-minute averages of the velocity field"... What do you mean by this? Not clear why 5 min averages should be used; seems an incorrect definition of turbulent flux

Thank you for this question. We used different window length to calculate turbulence statistics and found they asymptote when using window length of at least 20 and 10 minutes for the U12-C0.3 and U12-C0.5 simulations, respectively. We re-calculated the fluxes and variances using 20-minute time windows. Furthermore, we modified the manuscript and all the relevant figures accordingly.

[Figure]

Figure xv: Horizontal and vertical velocity variance as a function of the window length used to calculate the perturbations from the mean velocity. The left and right panels represent each atmospheric condition.

- Figure 12: please use confidence intervals rather than deciding for the reader what is significant and what not

Thank you for this suggestion. We considered including confidence intervals in our figure but restrained from doing so in the manuscript. We prefer to include all our results in one figure for the reader to compare the different stability cases and wind plant-single turbine cases. Adding confidence intervals for each line adds too much confusion to the figure, as shown in the figure below:

[Figure]

Figure xvi: Normalized wind speed deficit upstream of an isolated turbine (dashed lines) and first row of turbines in a wind plant (solid lines). The errorbars represent the 95% confidence level of the normalized velocity deficit.

We rather edited the manuscript to include a definition of statistical significance for readers unfamiliar with this topic:

L180-181: "*From this point onward, a statistically significant velocity deficit is such that its 95% confidence interval does not contain the value of 0 $\left( \overline{U}_{def} \pm CI \notin 0 \right)$.*"

- page 17, line 336: "A spectral analysis on the vertical and horizontal velocity at multiple locations in our simulations shows no statistical significant evidence of waves moving through our domain." What do you mean with waves moving through the domain? Do you mean that you did a frequency analysis? That does not make sense, since the gravity waves would be stationary…

Thank you for pointing that our description of the spectral analysis was inaccurate. We performed a spectral analysis on the horizontal velocity at multiple y-locations in our domain. As such, we find the dominant wavenumbers for the flow in the streamwise direction. The hub-height velocity spectra averaged over the y-direction (over the y-grid cells that contain turbines) shows the turbines only introduce power at a wavenumber of $k_x \sim 10^{-3}\ m^{-1}$, which is of the same order as the turbine spacing in the streamwise direction. Furthermore, both the simulations with and without the turbines display the maximum power at a wavenumber of $k_x \sim 8 \times 10^{-5}\ m^{-1}$ suggesting the turbines do not instigate standing waves throughout the domain, but rather the evolution of TKE throughout the domain explains the low-wavenumber oscillation in hub-height wind speed (See Major Comment #1) .

[Figure]

Figure xvii: Normalized energy spectrum as a function of wavenumber for the simulations with and without the wind plant for the U12-C0.3 (left) and U12-C0.5 (right) case.

The manuscript is modified as follows:

L364-365: *"We performed a spectral analysis on the spatial evolution of the horizontal velocity and found no evidence of standing waves in our simulations."*

**References**

Banta, R. M.: Stable-boundary-layer regimes from the perspective of the low-level jet, Acta Geophys., 56, 58–87, https://doi.org/10.2478/s11600-007-0049-8, 2008.

Banta, R. M., Newsom, R. K., Lundquist, J. K., Pichugina, Y. L., Coulter, R. L., and Mahrt, L.: Nocturnal Low-Level Jet Characteristics Over Kansas During Cases-99, Boundary-Layer Meteorology, 105, 221–252, https://doi.org/10.1023/A:1019992330866, 2002.

Banta, R. M., Pichugina, Y. L., and Brewer, W. A.: Turbulent Velocity-Variance Profiles in the Stable Boundary Layer Generated by a Nocturnal Low-Level Jet, 63, 2700–2719, https://doi.org/10.1175/JAS3776.1, 2006.

Bonin, T. A., Klein, P. M., and Chilson, P. B.: Contrasting Characteristics and Evolution of Southerly Low-Level Jets During Different Boundary-Layer Regimes, Boundary-Layer Meteorol, 174, 179–202, https://doi.org/10.1007/s10546-019-00481-0, 2020.

Conangla, L. and Cuxart, J.: On the Turbulence in the Upper Part of the Low-Level Jet: An Experimental and Numerical Study, Boundary-Layer Meteorol, 118, 379–400, https://doi.org/10.1007/s10546-005-0608-y, 2006.

Karipot, A., Leclerc, M. Y., Zhang, G., Lewin, K. F., Nagy, J., Hendrey, G. R., and Starr, G.: Influence of nocturnal low-level jet on turbulence structure and $CO_2$ flux measurements over a forest canopy, J. Geophys. Res., 113, D10102, https://doi.org/10.1029/2007JD009149, 2008.

Klemp, J. B. and Lilly, D. K.: Numerical Simulation of Hydrostatic Mountain Waves, 35, 78–107, https://doi.org/10.1175/1520-0469(1978)035<0078:NSOHMW>2.0.CO;2, 1978.

Mahrt, L. and Vickers, D.: Contrasting vertical structures of nocturnal boundary layers, Boundary-Layer Meteorology, 105, 351–363, https://doi.org/10.1023/A:1019964720989, 2002.

Mirocha, J. D., Churchfield, M. J., Muñoz-Esparza, D., Rai, R. K., Feng, Y., Kosović, B., Haupt, S. E., Brown, B., Ennis, B. L., Draxl, C., Sanz Rodrigo, J., Shaw, W. J., Berg, L. K., Moriarty, P. J., Linn, R. R., Kotamarthi, V. R., Balakrishnan, R., Cline, J. W., Robinson, M. C., and Ananthan, S.: Large-eddy simulation sensitivities to variations of configuration and forcing parameters in canonical boundary-layer flows for wind energy applications, Wind Energ. Sci., 3, 589–613, https://doi.org/10.5194/wes-3-589-2018, 2018.

Moeng, C.-H. and Sullivan, P. P.: A Comparison of Shear- and Buoyancy-Driven Planetary Boundary Layer Flows, 51, 999–1022, 1994.

Moeng, C.-H., Dudhia, J., Klemp, J., and Sullivan, P.: Examining Two-Way Grid Nesting for Large Eddy Simulation of the PBL Using the WRF Model, 135, 2295–2311, https://doi.org/10.1175/MWR3406.1, 2007.

Muñoz-Esparza, D. and Kosović, B.: Generation of Inflow Turbulence in Large-Eddy Simulations of Nonneutral Atmospheric Boundary Layers with the Cell Perturbation Method, Mon. Wea. Rev., 146, 1889–1909, https://doi.org/10.1175/MWR-D-18-0077.1, 2018.

Muñoz-Esparza, D., Kosović, B., Mirocha, J., and van Beeck, J.: Bridging the Transition from Mesoscale to Microscale Turbulence in Numerical Weather Prediction Models, Boundary-Layer Meteorol, 153, 409–440, https://doi.org/10.1007/s10546-014-9956-9, 2014.

Muñoz-Esparza, D., Kosović, B., van Beeck, J., and Mirocha, J.: A stochastic perturbation method to generate inflow turbulence in large-eddy simulation models: Application to neutrally stratified atmospheric boundary layers, Physics of Fluids, 27, 035102, https://doi.org/10.1063/1.4913572, 2015.

Peña, A., Kosović, B., and Mirocha, J. D.: Evaluation of idealized large-eddy simulations performed with the Weather Research and Forecasting model using turbulence measurements from a 250 m meteorological mast, Wind Energ. Sci., 6, 645–661, https://doi.org/10.5194/wes-6-645-2021, 2021.

Saiki, E. M., Moeng, C.-H., and Sullivan, P. P.: Large-Eddy Simulation Of The Stably Stratified Planetary Boundary Layer, Boundary-Layer Meteorology, 95, 1–30, https://doi.org/10.1023/A:1002428223156, 2000.

Wang, Y., Klipp, C. L., Garvey, D. M., Ligon, D. A., Williamson, C. C., Chang, S. S., Newsom, R. K., and Calhoun, R.: Nocturnal Low-Level-Jet-Dominated Atmospheric Boundary Layer Observed by a Doppler Lidar over Oklahoma City during JU2003, 46, 2098–2109, https://doi.org/10.1175/2006JAMC1283.1, 2007.

---

## Author Comment (AC2)

**Anonymous Reviewer 2:**

Dear Anonymous Reviewer,

Thank you for your feedback. It helped us to improve the manuscript and strengthen our findings. You emphasized some central points that were overlooked in our initial manuscript.

All reviewer comments appear in grey below, while authors' responses appear in blue text. Line numbers referenced in the authors' responses refer to the revised document. Figures included in the manuscript are labeled in italic and using numbers (e.g. *Figure 7*), while figures that only appear in the response to reviewer comments are labeled in smaller font and using roman numerals (e.g. Figure iv).

**General Comments**

The article 'Quantifying wind plant blockage under stable atmospheric conditions' by Gomez et al. draws conclusions about the magnitude and the sources of an observed velocity reduction upwind (blockage) of an idealized wind farm in two LES wind farm simulations. The main message of the article is that the blockage is higher in a stronger stratified atmospheric boundary layer and that the reason for this is the lack of vertical turbulent momentum transport. The authors further compare different virtual measurement setups to measure the wind speed upwind of the farm and analyse how a signal of blockage can be recognized in the production data. I see this work in general as an interesting addition to the current scientific discussion, but I see a couple of points that need to be addressed before I can recommend the full publication of the work.

**Major Comments**

1. Introduction: L. 26 - 35 - Definition of blockage, numbers

I have an issue with the introduction of blockage in wind farms. I would say it is not proven that the upstream wind speed necessarily decreases more when turbines are combined in a wind farm. Furthermore, I don't know any credible scientific publication that can relate the observed overpredictions of energy production of wind farms to blockage. The announcement of Orsted does not serve as a credible and sufficient reference. The numbers related to wake deficits (10 %) and blockage (1%) are not explained what they relate to. (wind speed? energy production?) If no reference is found here, I would suggest to rather talk about different orders of magnitude.

We appreciate your comment and include more references to complement our statements. Regarding the difference in blockage for wind plants and single turbines, we included references as follows:

L26-L27: "*And, when multiple turbines are combined into an array, the upstream wind speed decrease is larger than that of a turbine in isolation (Ebenhoch et al., 2017; Bleeg et al., 2018).*"

For the numbers related to wake deficits and blockage, we now include references to lidar observations of wakes and blockage that support these orders of magnitude for the wind speed deficit:

L31-32: "*Though wind speed deficits from wakes are large (~10%) (e.g. Lee and Lundquist, 2017), wind plant blockage produces wind speed deficits of ~1% over a wide area upstream (Schneemann et al., 2021), making it much more difficult to quantify.*"

Finally, we modify the statement that refers to Ørsted's announcement to suggest rather than state that there are overpredictions:

L29-30: "However, upstream wind plant blockage is usually neglected, possibly resulting in lower-than-forecasted energy predictions and financial losses for wind plant operators (Ørsted, 2019)."

2. Duration of averaging and influence on results and conclusions

The averaging period of 45 min appears quite small. If no longer averaging is possible, the limits of this restriction should be discussed throughout the paper. The paper draws conclusion on the significance of the results based also on the number of samples. For higher turbulence as in the lower stratification case, the significance is automatically lower. Thus, any conclusions about the significance of velocity differences, e.g. Figure 5, should point out that the significance criteria is strongly dependent on the length of measurement period. All conclusions about the significance of the results (wind speed deficit, power measurements) should relate to the selected sampling frequency (0.1 Hz) and the measurement period (45 min). A proper way to make the results between the two boundary layers more comparable is to scale the period of measurement to the turbulence level of the flow.

Thank you for raising this question that we had not yet considered this in our analysis. We now include this restriction in our manuscript as follows:

L177-180: "*Note that we use the same averaging time for two boundary layers that contain different turbulence levels. As such, our analysis automatically reduces the statistical significance of the U12-C0.3 simulations. Turbulence levels in the U12-C0.3 simulation are higher than in the U12-C0.5 simulations, resulting in higher correlation between the data and thus smaller independent sample size.*"

3. Discussion of measurement strategies (chapter 5)

I have a hard time grasping the meaning of and the approach in this chapter. I understand the conclusion is that more measurement points (and thus more samples) reduce the uncertainty, which is I would say common sense. So, in this case it would be more interesting to look at the combination of different sampling frequencies and locations. Also, I don't understand why uncertainty is not displayed to evaluate the measurement setups, but rather a bias. Furthermore, I don't think the averaging setups are even supposed to result into the same free stream velocity, as it can be clearly seen in Figures 3 and 9 that the flow is highly inhomogeneous in x. In consequence my suggestion would be to either remove the chapter or put a lot more effort in working out the implications of the different measurement setups.

Thank you for this constructive comment. This chapter is intended to point to the difficulties of measuring blockage in experimental setups. The different methodologies we employ to define the freestream velocity are not supposed to result in the same freestream velocity because of the cross-stream and streamwise fluctuations in the flow. However, they point to the difficulties of capturing the blockage effect using point measurements. We changed the nomenclature throughout the section, added an uncertainty analysis and highlighted our findings in a clearer way.

We modified how we reference each measurement methodology as follows:

L256-263: "*After removing the background flow from the velocity field, we calculate the freestream velocity upstream of the wind plant in multiple ways. We test five different approaches as shown in Fig. 10: 1) time-averaged, hub-height wind speed measured at one point 10D upstream of the wind plant $\left(\overline{U}_{\infty 1PM}\right)$, such as would be available from a single profiling lidar or meteorological tower; 2) time-averaged, hub-height wind speed measured at three points 10D upstream of the wind plant $\left(\overline{U}_{\infty 3PM}\right)$; 3) time-averaged, hub-height wind speed measured at six points 10D and 20D upstream of the wind plant $\left(\overline{U}_{\infty 6PM}\right)$; 4) time- and spatially averaged hub-height wind speed measured over the area extending 1D to 20D upstream of the wind plant $\left(\overline{U}_{\infty A}\right)$, such as would*

*be available from a scanning lidar; and 5) time- and spatially averaged hub-height wind speed measured over the whole turbulent domain of the no-turbine simulations (referred to as "True freestream" $\overline{U}_{\infty_T}$)."*

Furthermore, the convention for $\overline{U}_{\infty_i}$ can be directly related to Figure 10 and Figure 12.

*"Figure 10:  Schematic showing the relative location of the wind plant and the sampling locations for defining the freestream velocity of the flow. The freestream velocity in (a), (b), and (c) is calculated using one-, three-, and six-point measurements (PM), respectively. In (d) and (e) the freestream velocity is calculated from areal measurements enclosed by the dashed yellow line. The solid vertical lines represent the individual wind turbines. As such panel (e) represents the simulation with no turbines in the domain. The black crosses represent the locations for sampling the freestream velocity using point measurements. Freestream velocities in each panel are color-coded for each stability condition: red (blue) text represents the U12-C0.3 (U12-C0.5) case.*

[Figure]

For the uncertainty analysis, we include the new Figure 11 and Table 3. The manuscript is modified as follows:

L269-274: *"Depending on how the freestream velocity is defined, cross-stream flow inhomogeneities produce variations in the freestream velocity comparable in magnitude to the blockage effect (Figure 11). For the U12-C0.3 case, the difference between the freestream velocity estimated with a single point measurement and sampling the area upstream of the wind plant is 0.3 m s$^{-1}$, which is the same order of magnitude as the blockage effect we are trying to measure ($\widehat{\overline{U}}_{def} \sim 1\%$). For the U12-C0.5 case, the difference in the various definitions of the freestream velocity is smaller relative to the blockage effect but still present. The largest difference between freestream velocities is 1%."*

*"Figure 11: Uncertainty in estimating the freestream velocity using the methodologies outlined in Fig. 10. Errorbars represent the 95% confidence intervals for each measurement methodology.*

[Figure]

"

"*Table 3: Furthest distance upstream that displays a statistically significant velocity deficit for each $\overline{U}_{\infty i}$. Statistical significance is evaluated using a 95\% confidence level.*

| | Induction zone for each $\overline{U}_{\infty i}$ | | | | |
|---|---|---|---|---|---|
| *Case* | $\overline{U}_{\infty_{1PM}}[D]$ | $\overline{U}_{\infty_{3PM}}[D]$ | $\overline{U}_{\infty_{6PM}}[D]$ | $\overline{U}_{\infty_A}[D]$ | $\overline{U}_{\infty_T}[D]$ |
| *U12-C0.3* | *-18.5* | *-18* | *-6.5* | *-6.5* | *-4* |
| *U12-C0.5* | *-10* | *-20* | *-20* | *-13.5* | *-13* |

"

L286-292: "*We evaluate the statistical significance of the velocity deficit upstream of the wind plant for each $\overline{U}_{\infty i}$. The furthest distance upstream of the wind plant that displays a statistically significant velocity deficit changes with definition of freestream velocity (Table 3). Generally, the induction zone shrinks as the number of observations used to define the freestream velocity increases. Stronger cross-stream flow inhomogeneities in U12-C0.3 result in larger differences in the induction zone for the different methodologies used to define $\overline{U}_{\infty i}$ compared to U12-C0.5. Though a smaller density of observations increases the uncertainty in estimating the induction zone of a wind plant, the differences in the induction zone extent are primarily affected by the change in magnitude of the freestream velocity.*"

We do not include the confidence intervals in the velocity deficit because they will add too much clutter to the plot. This would require adding error bars to five different line plots in one same panel. We now make this clear in the caption for this figure in the manuscript:

"*Figure 12: Normalized velocity deficit for the (a) U12-C0.3 and (b) U12-C0.5 case using the various definitions of freestream velocity shown in Fig. 10. Results for the True freestream velocities are color-coded for each stability condition: red (blue) text represents the U12-C0.3 (U12-C0.5) case. Note that the colored lines in (a) and (b) are not the same as the corresponding lines in Figure 6 because here the freestream is single-valued, whereas in Figure 6 the freestream varies in the streamwise direction. Confidence intervals are not shown.*"

[Figure]

4. Conclusion on difference between the two simulations derived from flux divergence

> I suggest adding the flux divergences from the simulations without any wind farms. As the flow does not appear to be stationary along x, I would assume that there is already divergence even without any wind farm. Like this I am still a bit skeptic to accept the difference in the vertical momentum flux to be the sole reason for the difference in upstream wind speed deficit. Also, what about the mean momentum fluxes?

We appreciate this comment as we agree other terms are also very relevant. We analyze the following terms, and how they differ from the simulations without the GAD: $\overline{u}_i \frac{\partial \overline{u}}{\partial x_i}, \frac{1}{\rho}\frac{\partial \overline{p}}{\partial x}$, and $\frac{\partial \overline{u'u_i'}}{\partial x_i}$.

The induction zone seems to be most impacted by $\overline{w}\frac{\partial \overline{u}}{\partial z}$. The dominant terms in the x-momentum equation are $\overline{u}\frac{\partial \overline{u}}{\partial x}, \overline{w}\frac{\partial \overline{u}}{\partial z}$, and $\frac{1}{\rho}\frac{\partial \overline{p}}{\partial x}$. The pressure gradient term $\left(\frac{1}{\rho}\frac{\partial \overline{p}}{\partial x}\right)$ is nearly identical for both simulations, thus the difference in the wind plant's induction zone is largely determined by the advection of x-momentum by the vertical velocity $\left(\overline{w}\frac{\partial \overline{u}}{\partial z}\right)$. The negative vertical transport of x-momentum across the rotor layer is larger for the -0.5 K h$^{-1}$ simulation compared to the -0.3 K h$^{-1}$ simulation due to stronger vertical shear of the horizontal velocity. This larger negative vertical transport of x-momentum in turn requires a larger positive streamwise transport of x-momentum for the -0.5 K h$^{-1}$ simulation compared to the -0.3 K h$^{-1}$ simulation. We modified the manuscript as follows:

L203-219: "*In evaluating the x-momentum equation, we assume the flow is steady. This is a fair assumption because the cooling rate at the surface, which drives unsteadiness in the flow, results in small changes over the 40-min time averaging period. Furthermore, we neglect the Coriolis force in this analysis because the domain size (12000 m) is small compared to the scales affected by the Earth's rotation L=U/f=$\mathcal{O}(10^5\ m)$. In such a way, we consider momentum advection by the mean flow, pressure divergence, and the divergence of turbulent momentum fluxes. The turbulent fluxes are calculated from 20-minute averages of the velocity fields.*

*Figure 7: Mean flow momentum advection (a), and pressure gradient (b) terms of the x-momentum equation averaged spatially across the wind plant (y-direction) and the turbine rotor layer (z-direction). The plots show the departure of the terms in the momentum equation from the flow without the GAD.*

[Figure]

*The induction zone of the wind plant is most affected by the vertical transport of zonal momentum across the rotor layer (Figure 7). The pressure gradient term remains nearly equal for the U12-C0.5 and U12-C0.3 simulations (Figure 7b), given that the drag exerted by the turbines on the flow is very similar for both cases. Conversely, the negative vertical transport of x-momentum across the rotor layer is larger for U12-C0.5 compared to U12-C0.3 due to stronger vertical shear of the horizontal velocity (solid line in Figure 7a). As the flow is forced to move above the wind plant, the vertical momentum transport is balanced by the streamwise momentum advection. The larger vertical momentum loss in U12-C0.5 compared to U12-C0.3 requires additional streamwise advection of x-momentum for the flow to remain steady (Figure 7a), producing more flow deceleration up to 10D upwind of the wind plant. Turbulence divergence plays a minor role in the region upwind of the wind plant (not shown). Though the turbulence divergence terms are larger for U12-C0.3 compared to U12-C0.5, these remain virtually unchanged for the simulation with and without the GAD, suggesting they do not contribute significantly to momentum replenishment upwind of the turbines."*

We no longer consider turbulent momentum fluxes to play a major role in the induction zone of the wind plant. The turbulent momentum flux divergence terms still act to replenish momentum in the U12-C0.3 simulation, however, these terms are one order of magnitude smaller than the mean flow momentum transport and pressure divergence terms. Furthermore, we compare these terms for the simulations with and without the GAD (solid and dashed lines in figure below) and there is virtually no difference for both simulations. This implies turbulent momentum fluxes do not influence the induction zone of the wind plant.

[Figure]

Figure i: Turbulence momentum flux divergence terms for the U12-C0.3 (red) and U12-C0.5 (blue) simulations. The solid and dashed lines represent the fluxes for the simulation with and without the GAD, respectively.

We also modified the discussion section accordingly and removed the discussion on turbulent momentum fluxes:

*L337-339: "A highly stratified atmosphere hinders turbulent motions, increasing vertical shear of the horizontal velocity and thus modifying mean momentum advection across the rotor layer."*

**Minor Comments**

L 1: It's not true that only the first row of the plant is influenced

Thank you for pointing out this imprecision, we modified Abstract and Discussion sections accordingly.

L 95: *with a smaller time step*

The manuscript was modified accordingly.

Table 1: should also have the height of the two domains

Thank you for pointing this out. We now include the domain size in the vertical direction as well.

Figure 3: The graph looks like a much longer domain would be necessary to derive at a quasi-stationary region along x. Were any sensitivity studies done for the choice of the simulation domain?

It is true that turbulence is not completely stationary in our domain. However, it remains close to stationary in the region of interest. We modified this section on turbulence as follows:

L147-154: "*We determine whether turbulence has propagated throughout the entire domain using the variance of the vertical velocity, which is calculated using 20 min time windows. Turbulence is close to steady 20 min after initializing the nested domain for U12-C0.3 (Figure 3a), and 30 min after initializing the nested domain for U12-C0.5 (Figure 3b), and results are discarded before these times. Furthermore, turbulence in the surface layer becomes quasi-stationary after x = 4000 m for both simulations (Figure 4), and we also discard results upstream of this location. Although we trigger turbulent motions up to the capping inversion, turbulence in the residual layer decays rapidly throughout the domain and becomes small after x = 4000 m. Note that minimal turbulence persists in the residual layer, as is sometimes observed in regions with flat terrain (Banta et al., 2006; Banta, 2008; Bonin et al., 2020).*

*Figure 3: Time evolution of the vertical velocity variance for the U12-C0.3 (a), and U12-C0.5 (b) simulation without the wind turbines. The profiles are averaged over a 0.8 km² region centered at the first row of the wind plant (x = 6804 m, y = 5890 m). The perturbations of the vertical velocity are calculated using a 20-min moving average.*

[Figure]

*Figure 4: Evolution of the vertical velocity variance averaged in the y-direction across the domain for the U12-C0.3 (a), and U12-C0.5 (b) simulations without the turbines. Vertical profiles are color coded for each x-location in the domain and plotted in 1000 m increments.*

[Figure]

We discuss the repercussion of this non-stationarity in section 5:

L241– L246: *"Our simulations also display some variability in the streamwise "background" flow (Figure 9b). The streamwise variability in the hub-height horizontal velocity results from turbulence development throughout the domain. As turbulent motions develop throughout the domain, higher momentum is transported downwards across the rotor layer, increasing horizontal velocity at hub height. This downward transport of momentum and turbulence kinetic energy is sometimes observed in stable boundary layers with low-level jets (Karipot et al., 2008; Banta et al., 2002; Mahrt and Vickers, 2002; Conangla and Cuxart, 2006; Wang et al., 2007)."*

We also analyzed in more detail the relationship between turbulence and horizontal velocity. We perform a principal component analysis on the time-averaged horizontal velocity, and TKE across the domain for the simulations without the GAD parameterization. We evaluate the spatial pattern of each variable in the $x$- and $z$-directions for the U12-C0.3 simulation, which displays the largest streamwise variations in horizontal velocity. The dominant spatial pattern for the evolution of TKE in the streamwise direction matches a dominant spatial pattern for the evolution of the horizontal velocity. The correlation in the empirical orthogonal functions of hub-height horizontal wind speed and TKE along the $x$-direction is 0.75 for the U12-C0.3 simulation. The empirical orthogonal functions for TKE and horizontal wind speeds at $y = 5200\ m$ that explain 24% and 18% of variance, respectively, for the U12-C0.3 simulation are shown below.

[Figure]

Figure ii: Empirical orthogonal functions for turbulence kinetic energy (top) and horizontal wind speed (bottom) at the center of the domain in the y-direction.

The empirical orthogonal functions demonstrate a downward vertical transport of turbulence kinetic energy and momentum, which result in an increase in hub-height wind speeds. It is not uncommon for stable boundary layers over land to display a downward transport of momentum in the presence of a low-level jet (Karipot et al., 2008; Banta et al., 2002; Mahrt and Vickers, 2002; Conangla and Cuxart, 2006; Wang et al., 2007), as indeed occurs in both of our experimental setups.

Figure 5: Why are the lines not converging to zero at 20 D?

We appreciate this comment and we added clarification in the manuscript. Our simulations show there is indeed an effect out to 20D, however, the effect is not statistically significant (95% confidence level). We define the induction zone using a statistical analysis on the velocity fields. Therefore, although other studies state the wind plant influences the velocity field upstream all the way to the domain inflow boundary (e.g. Wu and Porté-Agel, 2017; Allaerts and Meyers, 2018) based on the mean velocity, we do not consider the induction zone extends that far upstream because our statistical analysis suggests this small deviation is not statistically significant.

The figure below shows the confidence intervals on the normalized velocity deficit from the strong stably stratified simulation. This shows the mean velocity deficit is slightly less than zero at 20D, but the confidence intervals are not below zero. Therefore, the induction zone as we define it does not extend beyond 20D. Confidence intervals are not included in Figure 6 in the manuscript because we want to aggregate the results from the different cases (i.e. different stability cases, and single turbine and wind plant) into one same plot, while showing the statistical significance of our results.

[Figure]

Figure iii: Normalized velocity deficit for the U12-C0.5 case. The error bars represent the 95% confidence intervals.

The manuscript is modified as follows:

L168-169: *"We define the induction zone with the statistical significance of the velocity deficit upstream of the first row of turbines in the plant."*

L186-188: *"The wind plant modifies the flow in a statistically significant manner up to 15D upstream for U12-C0.5. Conversely, there is not enough statistical evidence that the induction zone extends further than 2D upstream for U12-C0.3."*

L346-347: *"We do not find statistical evidence of a far-reaching induction zone for the U12-C0.3 case. For this weaker stable layer, the velocity deficit is only statistically significant up to 2.5D upstream of the wind plant."*

Figure 8b: From Figure 8a I would assume that the difference between the inversion height upstream and downstream should be a lot higher in the strongly stratified case than displayed here.

Thank you for catching this. We were plotting the potential temperature profiles at the wrong location downstream. We modified the figure to include the correct locations downstream. Please see revised Fig. 8.

L 289: See comment for L1

Thank you for pointing out this imprecision, we modified Abstract and Discussion sections accordingly.

Discussion & Conclusions: For me the chapter is too long and hard to read. I suggest restructuring the chapter. The part of the non-existent gravity waves for example could be written much shorter and more concise.

Thank you for this comment, we modified the discussion section to make it more concise and clearer.